# Counting Like Transformers: Compiling Temporal Counting Logic Into Softmax Transformers

**Andy Yang & David Chiang**
University of Notre Dame
{ayang4,dchiang}@nd.edu

## Abstract

Deriving formal bounds on the expressivity of transformers, as well as studying transformers that are constructed to implement known algorithms, are both effective methods for better understanding the computational power of transformers. Towards both ends, we introduce the temporal counting logic $\mathbf{K}_t[\#]$ alongside the RASP variant **C-RASP**. We show they are equivalent to each other, and that together they are the best-known lower bound on the formal expressivity of future-masked soft attention transformers with unbounded input size. We prove this by showing all $\mathbf{K}_t[\#]$ formulas can be compiled into these transformers without any additional positional embeddings.

## 1 Introduction

What problems can transformers (Vaswani et al., 2017) solve, what problems can they not solve, and how can we prove it? Formal logic, in connection with programming language theory, formal language theory, and finite model theory, give a framework in which to investigate these questions.

Previous theoretical work, as surveyed by Strobl et al. (2024), has advanced our understanding of transformers immensely. However, it has not provided a full account of their expressive power. Much of this work only considers modifications of transformers, like average-hard attention transformers (AHATs) (Barceló et al., 2024) or unique-hard attention transformers (UHATs) (Angluin et al., 2023), which are not known to be either a subset or superset of standard, soft-attention transformers (SMATs). At the same time, programming languages like RASP (Weiss et al., 2021) propose a human-readable language with which to understand transformer computations. However, current languages compile into transformers that appear to be more powerful than standard transformers (Weiss et al., 2021), or require approximations and restrictions on input length (Lindner et al., 2023) to do so.

Here, we target soft attention transformers (as originally defined (Vaswani et al., 2017) and as used in practice). We prove that future-masked soft attention transformer encoders, with no restriction on input length, can recognize all the formal languages defined by formulas of $\mathbf{K}_t[\#]$, a temporal counting logic. Along the way we develop a RASP variant called **C-RASP**, equivalent to $\mathbf{K}_t[\#]$. Both are, to our knowledge, the tightest-known lower bound on the expressivity of soft attention transformer encoders. Our contributions are as follows:

- We define **C-RASP**, the first variant of RASP that provably compiles into future-masked soft attention transformer encoders with no restrictions on the input length.
- We prove that **C-RASP** is equivalent to $\mathbf{K}_t[\#]$.
- We prove that the previous best lower bound, FOC[+; MOD] (Chiang et al., 2023), is strictly less expressive than $\mathbf{K}_t[\#]$.
- We prove that transformers which use fixed-precision numbers (as real-world transformers do) can be compiled back to $\mathbf{K}_t[\#]$.

## 2 Background

Previous theoretical work on the expressivity of transformers has related them to a variety of automata, circuit classes, and logics, all under varying assumptions (Strobl et al., 2024). Here, we focus on using logics to characterize soft attention transformer encoders with future masking. In particular, these encoders perform the same computations as one step of a transformer decoder. Decoder-only models like GPT (OpenAI, 2023) currently dominate applications of transformers, while empirical work has noted they have significant limits and perplexing behavior. We believe theoretical analysis can provide valuable insight towards understanding how to best use these models in practice.

The previous best upper bound on log-precision transformers (which are argued to closely approximate real-life behavior) is $\textbf{TC}^0$ (Merrill and Sabharwal, 2023). The previous best lower bound is $\mathrm{FOC}[+; \mathrm{MOD}]$ (Chiang et al., 2023). We strengthen this lower bound using $\mathbf{K}_t[\#]$, a temporal counting logic, and **C-RASP**, a new variant of the programming language RASP. We show both can be simulated by soft attention transformers.

### 2.1 Hard and Soft Attention

Many previous works have investigated the expressivity of hard attention transformers, including Angluin et al. (2023); Barceló et al. (2024); Yao et al. (2021). However, reconciling the differences between the theoretical model of AHATs and the standard SMATs actually remains a very open area of inquiry at the moment.

In particular, it is not yet clear how the expressivity bounds on average-hard attention transformers apply to softmax attention transformers. On the one hand, average-hard attention transformers can express sparse attention patterns by assigning weights of zero to positions, but soft attention transformers can only approximate these patterns (scores are always non-zero). On the other hand SMATs can express non-uniform attention patterns, which AHATs cannot. As such it is not known whether softmax attention can simulate average-hard attention (or even the other way around), so lower bounds on the expressivity of average-hard attention transformers cannot directly be applied to soft attention transformers.

Thus, while the counting logic $\mathbf{LTL}(\mathrm{C}, +)$ (Barceló et al., 2024) contains $\mathbf{K}_t[\#]$, we note that they only prove it is a lower bound for average-hard attention transformers, not soft attention transformers. We can view $\mathbf{K}_t[\#]$ and **C-RASP** as a gentle enough restriction of $\mathbf{LTL}(\mathrm{C}, +)$ so as to render it simulatable by softmax attention transformers, but still above previous known bounds. We hope future work will clarify the disparities between average-hard and softmax attention.

### 2.2 RASP and Tracr

Implementing algorithms in transformers using human-readable programming languages gives researchers and engineers a deeper understanding of the computations transformers can perform. We believe that using this formalism to understand transformers has not only pedagogical benefits, but theoretical ones as well. For example, this perspective has been used by Zhou et al. (2024) to shed light on the length-generalization capabilities of transformers.

These programming languages promise to compile into transformers that implement known algorithms, which are therefore interpretable by construction. However, existing examples make several unrealistic assumptions about transformers, rendering them inappropriate for compilation into standard transformers.

The primary example is RASP, which makes three strong assumptions. First, the transformers that RASP compiles into use average-hard attention, which are not known to be exactly simulated by soft attention transformers (although average-hard attention behavior has been observed to be learned approximately, in practice (Merrill et al., 2021)). Second, the attention weights (selectors) are not restricted to be dot-products of query and key vectors;

this allows compilation of expressions involving arbitrary binary predicates like $x = y$ or $x < y$. Third, they assume position-wise feed-forward networks can implement any computable function, with the rationale that any continuous function can be approximated by the universal approximation theorem (Hornik et al., 1989).

Building on RASP, Tracr (Lindner et al., 2023) compiles a subset of RASP into standard transformers. It compiles RASP selectors to dot-product attention, although this requires a syntactic restriction on selectors and a maximum string length, and it compiles RASP element-wise operations to ReLU FFNs, though only approximately.

Furthermore, neither of these consider layer normalization, which Brody et al. (2023) show contributes to the expressivity of transformers.

Here, we define a variant of RASP that has more restrictions, but that can be compiled exactly into a soft attention transformer encoder. Our variant, **C-RASP**, is based on the temporal counting logic $\mathbf{K}_t[\#]$.

### 2.3 FOC[+; MOD]

Counting logics are a rich area (van Benthem and Icard, 2023) of logic, whose connection with transformers has been noted by Chiang et al. (2023) and Barceló et al. (2024). In essence, uniform attention patterns – where attention is spread evenly across positions – can very naturally simulate counting terms. Chiang et al. (2023) define a variant of first-order logic with counting quantifiers, called FOC[+; MOD], and prove that, on the one hand, any sentence of FOC[+; MOD] can be translated into an equivalent soft attention transformer encoder, and on the other hand, any *fixed-precision* soft attention transformer encoder can be translated into an equivalent sentence of FOC[+; MOD].

However, FOC[+; MOD] seems somewhat underpowered. It has a normal form that uses only one quantifier alternation ($\exists x. \exists^{=x} p. \cdots$) and only one position variable. This means the equivalent transformer only has depth 2, and only uses the output of self-attention at one position. By considering an ordering on positions (and future-masking on the corresponding transformers) we derive a much better lower-bound result.

### 2.4 Temporal logic

A technical challenge when simulating variants of first-order logic with transformers is that a formula with $k$ free variables, each of which is interpreted as a position in from 1 to $n$, would seem to correspond to a tensor of $n^k$ values in the corresponding transformer, but transformers only have $n$ values at each layer and $n^2$ values in the attention weights.

Whereas FOC[+; MOD] avoids this difficulty by using a normal form with only one variable, Angluin et al. (2023) and Barceló et al. (2024) avoid it by relying on linear temporal logic.

Temporal logics (Gabbay et al., 1980) have been widely adopted as tools for the formal verification of state properties during the execution of programs over time. Intuitively, temporal logics can be used to formalize statements such as the following:

> Until the first train arrived at the gate, the bar remained lowered.
> My arm has been sore since Tuesday.
> At no point will the temperature go below zero.

More abstractly, we can also use the syntax of temporal logic to specify the occurance of symbols in a string $w$.

> Until the first symbol $t$, $w$ contains only $l$'s.
> Only the symbol $s$ has appeared since position 2 in $w$.
> At no position does $w$ contains a $z$.

We believe that temporal logics are a very appropriate specification language for thinking about masked self-attention. Firstly, the temporal accessibility relation – properties at time $i$

can only depend on times $j \leq i$ – provides a natural way to model future-masking in transformers. Secondly, the restricted use of variables in temporal logic corresponds closely to the computational processing resources of the standard transformer, where scores depend on only two positions. Finally, temporal logics are highly-utilized in the field of formal methods (Fisher, 2011), so solidifying this connection with transformers may allow more ideas to get shared across disciplines.

## 3 C-RASP

We follow in the footsteps of Weiss et al. (2021) to define a variant of RASP called **C-RASP**. The audience may find the syntax of **C-RASP** is easier to follow, so we present it first before defining $\mathbf{K}_t[\#]$ (although both are equivalent in the end). In proofs, we generally prefer the more compact syntax of $\mathbf{K}_t[\#]$, but for writing programs, we use **C-RASP**.

### 3.1 Definitions

**Definition 3.1 (C-RASP).** A **C-RASP** program is defined as a sequence $P_1, \ldots, P_n$ of **C-RASP** operations. There are two types of operations:

| **Boolean-Valued Operations** | |
| --- | --- |
| **Initial** | $P(i) := Q_a(i)$ for $a \in \Sigma$ |
| **Boolean** | $P(i) := \neg P_1(i)$ |
| | $P(i) := P_1(i) \wedge P_2(i)$ |
| **Comparison** | $P(i) := C_1(i) \leq C_2(i)$ |
| **Constant** | $P(i) := 1$ |

| **Count-Valued Operations** | |
| --- | --- |
| **Counting** | $C(i) := \#[j \leq i] \ P(j)$ |
| **Conditional** | $C(i) := P(i) \ ? \ C_1(i) : C_2(i)$ |
| **Addition** | $C(i) := C_1(i) + C_2(i)$ |
| **Subtraction** | $C(i) := C_1(i) - C_2(i)$ |
| **Min/Max** | $C(i) := \min(C_1(i), C_2(i))$ |
| | $C(i) := \max(C_1(i), C_2(i))$ |
| **Constant** | $C(i) := 1$ |

Counting operations count the positions $j \leq i$ such that $P(j)$ holds, returning the sum. We could also extend the syntax to use an expression $P(i, j)$ which depends on both $i$ and $j$; this can be transformed to $P(j)$ since our logic has only unary predicates, as shown in Lemma A.1. Conditional operations return $C_1$ if $P$ holds, and $C_2$ otherwise.

By convention, when using a **C-RASP** program to recognize languages, we use the value of the *last* operation, which must be Boolean-valued, at the last position, to determine acceptance. That is, if the program is run on input $w$ with length $n$, and the last operation is $D$, then we accept $w$ if and only if $D(n)$ is true.

**Example 3.2.** We present a program to recognize Dyck-1 as an example. More annotated examples can be found in Appendix A.2

$$C_((i) := \#[j \leq i] \ Q_((j) \qquad \text{The number of ( up to position } i$$
$$C_)(i) := \#[j \leq i] \ Q_)(j) \qquad \text{The number of ) up to position } i$$
$$V(i) := C_((i) < C_)(i) \qquad \textit{Violation}: \text{there are more ) than (}$$
$$C_V(i) := \#[j \leq i] \ V(j) \qquad \text{The number of } \textit{Violations}$$
$$M(i) := C_V(i) = 0 \qquad \textit{Matched}: \text{zero } \textit{Violations}$$
$$B(i) := C_((i) = C_)(i) \qquad \textit{Balanced}: \text{same number of ( and )}$$
$$D(i) := M(i) \wedge B(i) \qquad \text{String is } \textit{Matched} \text{ and } \textit{Balanced}$$

## 4   The Temporal Counting Logic $\mathbf{K}_t[\#]$

In this section, we define the temporal counting logic $\mathbf{K}_t[\#]$.

### 4.1   Definitions

The temporal logic we target here is the past fragment of the Minimal Tense Logic (Rescher and Urquhart, 2012), with counting terms (Barceló et al., 2024). It can also be thought of as a modal logic with arithmetic constraints, like that of Demri and Lugiez (2010), simply restricted to the setting where the structures are strings.

We present the syntax of $\mathbf{K}_t[\#]$ in Backus–Naur form:[1]

$$F ::= Q_a \mid \neg F \mid F \wedge F \mid C \leq C \mid \top$$
$$C ::= \#[F] \mid C + C \mid C - C \mid 1$$

In temporal logics, formulas and terms are written without arguments because they are always interpreted with respect to a structure at a specified position. In our setting, they are interpreted with respect to a string $w$ at a position $i$, where $w = w_1 w_2 \cdots w_n$ and $i \in [1, n]$. A count term $C$ is interpreted as an integer, written $C^{w,i}$ and defined as follows.

$$
\begin{aligned}
\#[F]^{w,i} &= |\{j \in [1,i] \mid w, j \vDash F\}| \\
(C_1 + C_2)^{w,i} &= C_1^{w,i} + C_2^{w,i} \\
(C_1 - C_2)^{w,i} &= C_1^{w,i} - C_2^{w,i} \\
1^{w,i} &= 1
\end{aligned}
$$

As syntactic sugar, we allow the use of any natural number as a constant, which implicitly is defined as a sum of 1's. Similarly, 0 can be defined as $\#[\neg\top]$, and $i$ as $\#[\top]$. Next, the interpretation of a formula $F$ at position $i$, written $w, i \vDash F$, defined as follows:

$$
\begin{aligned}
w, i \vDash Q_a &\iff w_i = a \\
w, i \vDash \neg F &\iff w, i \nvDash F \\
w, i \vDash F_1 \wedge F_2 &\iff w, i \vDash F_1 \text{ and } w, i \vDash F_2 \\
w, i \vDash C_1 \leq C_2 &\iff C_1^{w,i} \leq C_2^{w,i} \\
w, i \vDash \top &\qquad \text{is always the case}
\end{aligned}
$$

We say that a string $w$ with length $n$ *end-satisfies* $\phi$ a formula of $\mathbf{K}_t[\#]$ whenever $w, n \vDash \phi$ (Maler and Pnueli, 1990). The language defined by $\phi$ is the set of all strings end-satisfied by $\phi$. As a final note, whenever $w$ is implicit, we can write $F(i)$ which is True iff $w, i \vDash F$ and also write $C(i)$ which is equal to $C^{w,i}$.

### 4.2   Examples

Although an exact characterization of $\mathbf{K}_t[\#]$ is not currently known, we can see that it can define a variety of regular, context-free, and non-context-free languages.

---

[1]We pronounce $\mathbf{K}_t[\#]$ as "K-t-sharp". The logic $\mathbf{K}_t$ is E.J. Lemmon's minimal tense logic (Rescher and Urquhart, 2012), where $\mathbf{K}$ is the minimal modal logic, and the t refers to "tense", indicating that our structures are linear, like timelines or strings. Additionally, $\#$ is the modal counting operator, which is fairly standard notation in counting logics.

| Language | Formula |
|---|---|
| $a^*b^*$ | $\#[Q_a \wedge (\#[Q_b] \geq 1)] = 0$ |
| $a^*b^*a^*$ | $\#[Q_b \wedge \#[Q_a \wedge (\#[Q_b] \geq 1)] \geq 1] = 0$ |
| $a^n b^n c^n$ | $\#[Q_b \wedge (\#[Q_c] = 0)] = \#[Q_b]$ |
| | $\wedge \#[Q_a \wedge (\#[Q_b \vee Q_c] = 0)] = \#[Q_a]$ |
| | $\wedge \#[Q_a] = \#[Q_b] \wedge \#[Q_b] = \#[Q_c] \wedge \#[Q_c] = \#[Q_a]$ |
| Dyck-1 | $\left(\#\left[Q_{(}\right] = \#\left[Q_{)}\right]\right) \wedge \left(\#\left[\#\left[Q_{)}\right] > \#\left[Q_{(}\right]\right] = 0\right)$ |
| hello | $\#[\top] = 5 \wedge Q_o \wedge \#[Q_l \wedge \#[Q_e \wedge \#[Q_h] = 1] = 1] = 2$ |

It is of note that the context sensitive language $a^n b^n c^n$ has been observed to be learnable by transformers (Bhattamishra et al., 2020).

## 4.3 Modal Depth

**Definition 4.1.** The *modal depth* of a formula $\phi$ or term $C$, which we notate as $md(\phi)$, is the maximum level of nesting of # terms. That is,

$$md(Q_\sigma) = 0 \qquad md(1) = 0$$
$$md(\neg\phi) = md(\phi) \qquad md(\#[\phi]) = 1 + md(\phi)$$
$$md(\phi_1 \wedge \phi_2) = \max(md(\phi_1), md(\phi_2)) \qquad md(C_1 + C_2) = \max(md(C_1), md(C_2))$$
$$md(C_1 \leq C_2) = \max(md(C_1), md(C_2))$$

The following construction gives some intuition on the effect of modal depth.

**Lemma 4.2.** *For every string $s$ of length $n$, there exists a formula $\phi_a$ of modal depth $n$ such that $w \vDash \phi_a$ if and only if $w$ contains $s$ as a subsequence.*

*Proof.* Let $s = s_1 s_2 \cdots s_n$. Then define $\phi_s := \tau_n \geq 1$ where

$$\tau_1 := \#[Q_{s_1}]$$

$$\tau_{k+1} := \begin{cases} \#\left[Q_{s_{k+1}} \wedge \tau_k \geq 1\right] & s_k \neq s_{k+1} \\ \#\left[Q_{s_{k+1}} \wedge \tau_k \geq 2\right] & s_k = s_{k+1} \end{cases}$$

Verification is left as an exercise for the reader. $\square$

As a consequence of the above and Theorem 5.7, we see that masked soft attention transformers can recognize all the piecewise testable languages (Klíma and Polák, 2010), a subset of the star-free languages. Recall, however, that $\mathbf{K_t}[\#]$ can express much more: for example, the context sensitive language $a^n b^n c^n$. A comprehensive study of the expressive power of $\mathbf{K_t}[\#]$ and **C-RASP** (and the effect of modal depth on expressivity) would be informative, and is left for future work.

## 4.4 $\mathbf{K_t}[\#]$ and C-RASP

**C-RASP** may have a more convenient syntax to write programs in, but **C-RASP** programs and $\mathbf{K_t}[\#]$ formulas are exactly equivalent in expressivity.

**Theorem 4.3.** *A C-RASP program recognizes language L iff a $\mathbf{K_t}[\#]$ formula defines L. More precisely, given alphabet $\Sigma$, for any $\mathbf{K_t}[\#]$ formula $\phi$ there is a C-RASP program P such that $w \in \Sigma^*$ end-satisfies $\phi$ iff $w$ is accepted by P, and vice versa.*

*Proof.* See Appendix A.3. $\square$

# 5 From $\mathbf{K_t}[\#]$ to Masked-Attention Transformers

In this section, we show how to compile $\mathbf{K_t}[\#]$ into transformers. It would also be possible to translate directly from **C-RASP** to transformers; this would use fewer dimensions, but more layers.

## 5.1 Transformers

We assume familiarity with the transformer architecture (Vaswani et al., 2017), and only review the basic definitions particular to our setting. To simplify our analysis, we do not consider positional encodings at first, deferring them to Section 6.

**Definition 5.1.** (Word Embeddings) Let $w$ be an input string of length $n$ over a finite alphabet $\Sigma$. We prepend to $w$ a special symbol BOS, which we assume is not in $\Sigma$. We abbreviate $n + 1$ as $n'$. A word embedding with dimension $d$ over $\Sigma$ is a function $WE \colon \Sigma \cup \{\text{BOS}\} \to \mathbb{R}^d$ applied position-wise to a string of length $n$ to form a tensor in $\mathbb{R}^{d \times n'}$.

**Definition 5.2** (Transformer Block). A transformer block $B$, defined with a dimension $d$, specifies a function $B \colon \mathbb{R}^{d \times n'} \to \mathbb{R}^{d \times n'}$ that computes

$$B(A) = LN_2(FFN(A') + A')$$
$$A' = LN_1(SA(A) + A)$$

where $SA$ denotes a self-attention layer, by the standard definition (Vaswani et al., 2017), $FFN$ denotes a two layer feed-forward neural network with ReLU activations between the layers, and $LN_1$ and $LN_2$ denote position-wise applications of LayerNorm. This setup is commonly referred to as a "post-norm" block.

## 5.2 Overview of the translation

The input and output of a transformer block are tensors in $\mathbb{R}^{d \times n'}$. The resulting sequence of tensors across transformer blocks is sometimes referred to as the "residual stream" (Elhage et al., 2021). We store the values of each subformula or count term of a $\mathbf{K}_t[\#]$ formula in a different dimension of the residual stream.

Let $A \in \mathbb{R}^{d \times n'}$ be a tensor in the residual stream. Formulas $\phi_k$ are stored as two rows of $A$:

$$A_{2k-1:2k,*} = \begin{bmatrix} 1 & -2\phi_k(1)+1 & -2\phi_k(2)+1 & \cdots & -2\phi_k(n)+1 \\ -1 & +2\phi_k(1)-1 & +2\phi_k(2)-1 & \cdots & +2\phi_k(n)-1 \end{bmatrix}.$$

Similarly, count terms $C_k$ are stored as:

$$A_{2k-1:2k,*} = \begin{bmatrix} 0 & -\frac{C_k(1)}{2} & -\frac{C_k(2)}{3} & \cdots & -\frac{C_k(n)}{n'} \\ 0 & +\frac{C_k(1)}{2} & +\frac{C_k(2)}{3} & \cdots & +\frac{C_k(n)}{n'} \end{bmatrix}.$$

The division of $C(i)$ by $(i + 1)$ is a consequence of the fact that attention computes an average rather than a sum. Dealing with these divisions is a common feature of many transformer constructions. In contrast to other constructions that undo the divisions using nonstandard embeddings (Pérez et al., 2021; Barceló et al., 2024) or nonstandard versions of LayerNorm (Merrill and Sabharwal, 2024), our construction uses no position embeddings and only standard LayerNorm. A minor consequence of our handling of comparison (Appendix B.3) is that while Boolean values are preserved throughout the computation, integer values can get overwritten.

The reason for representing every value as two transformer activation values is to account for LayerNorm. It ensures that all feature vectors have zero mean, so LayerNorm only applies a position-wise scaling factor. When necessary, we describe how to use LayerNorm to remove this scaling factor in Appendix B.3.

Observe that for subformulas, our convention states the BOS position is always false. Consequently, for count terms, the BOS position is always 0. We can ensure this with a feed-forward layer.

**Lemma 5.3.** *Using the word embedding and a single feed-forward layer, we can set the* BOS *position to False, without disturbing the Boolean value at any other position.*

*Proof.* See Appendix B.1. □

## 5.3 Counting using masked uniform attention

Count terms in $\mathbf{K}_t[\#]$ can be simulated by uniform self-attention layers.

**Lemma 5.4.** *Let $A_{2k-1:2k},*$ store a Boolean vector as defined above. For any $i$, let $C_{k,i}$ be the number of positions $j \leq i$ such that $A_{2k-1:2k,j}$ is True. Then there is a transformer block that computes, at each position $i$, and in two other dimensions $2k'-1, 2k'$, the values $-\frac{C_{k,i}}{i+1}$ and $\frac{C_{k,i}}{i+1}$.*

*Proof.* See Appendix B.2 for proof and pictures. □

## 5.4 Other operations using position-wise feed-forward networks

All other $\mathbf{K}_t[\#]$ formulas and terms can be simulated by feed-forward layers.

**Lemma 5.5.** *The following position-wise operations can be simulated by a single transformer block, using existing dimensions as input and a fresh dimension as output: addition $(+)$, subtraction $(-)$, comparison $(\leq)$, and Boolean operations $(\wedge, \neg)$.*

*Proof.* See Appendix B.3. □

## 5.5 Compiling $\mathbf{K}_t[\#]$ formulas into masked uniform attention transformers

**Definition 5.6.** Fix an alphabet $\Sigma$, and assume that the symbol BOS is not in $\Sigma$. We say a masked soft attention transformer $T$ (as a composition of blocks $T = B_b \circ \ldots \circ B_1 \circ WE$) with $d$ dimensions *simulates* a $\mathbf{K}_t[\#]$ formula $\phi$ if for every input $w \in \Sigma^*$ with length $n$ and every subformula $\psi_k$ of $\phi$, there is some dimension $d_k$ such that

$$[T(\text{BOS} \cdot w)]_{2d_k-1:2d_k,i+1} = \begin{cases} \begin{bmatrix} -1 \\ +1 \end{bmatrix} & \text{if } w, i \vDash \psi_k \\ \\ \begin{bmatrix} +1 \\ -1 \end{bmatrix} & \text{otherwise.} \end{cases}$$

**Theorem 5.7.** *For every $\mathbf{K}_t[\#]$ formula $\phi$, there exists a soft attention transformer which simulates $\phi$. Moreover, the transformer will have at most $md(\phi)$ blocks.*

*Proof.* We induct on the modal depth of $\phi$. If $\phi$ is of modal depth 0, it must be a Boolean combination of $Q_\sigma$ formulas. This can be simulated in the word embedding.

For the inductive step, let $\phi$ be a $\mathbf{K}_t[\#]$ formula of modal depth $m + 1$. By Section 4.3, $\phi$ is a Boolean combination of:

– Subformulas of modal depth at most $m$.
– Subformulas of the form $\sum_{k \in K} a_k \#[\psi_k] \geq 0$, where $K$ is a set of indices, $a_k$ are integers, and $\psi_k$ are subformulas of modal depth $m$.

By the inductive hypothesis, for each subformula $\psi_k$ of modal depth at most $m$, there is a transformer $T_k$ which simulates it. Parallel-compose all the $T_k$ as described by Appendix B.4 into a single transformer. Then we need to perform the following operations in sequence:

1. Compute $\#[\psi_k]$ for all relevant $\psi_k$, as described in Lemma 5.4.
2. Compute all formulas of the form $\sum_{k \in K} a_k \#[\psi_k] \geq 0$, as described in Appendix B.3.
3. Compute all Boolean combinations of the above subformulas as necessary.
4. Ensure the BOS position is False.

This can be achieved by adding one block. The first step can be achieved with a self-attention layer. We've described how to compute each of the next three steps individually using a feed-forward layer, but their composition can also be performed with a single feed-forward layer. □

## 6 Relationship to other formalisms

The expressivity of $\mathbf{K}_t[\#]$ can be characterized in relation to other bounds on transformer formal expressivity. To begin with, the previous best lower bound on soft attention transformer encoders was $\text{FOC}[+; \text{MOD}]$, by Chiang et al. (2023). However, $\mathbf{K}_t[\#]$ forms a tighter lower bound due to its ability to model order.

**Lemma 6.1.** $\mathbf{K}_t[\#]$ *formulas can simulate* $\text{FOC}[+]$ *formulas, and* $\mathbf{K}_t[\#; \text{MOD}]$ *can simulate* $\text{FOC}[+; \text{MOD}]$. *In general, for any set* $\mathcal{P}$ *of unary predicates, the extension* $\text{FOC}[+; \mathcal{P}]$ *is contained in* $\mathbf{K}_t[\#; \mathcal{P}]$.

*Proof.* See Appendix A.3. □

We've seen that **C-RASP** can define Dyck-1 (Example 3.2), and Bhattamishra et al. (2020) prove that transformers, under the same assumption as ours, can express Dyck-1 via a very similar construction. However, it's easy to show that $\text{FOC}[+; \text{MOD}]$ cannot define Dyck-1. Thus, $\mathbf{K}_t[\#]$ is a more realistic lower bound for transformers than $\text{FOC}[+; \text{MOD}]$:

**Proposition 6.2.** *There is no sentence of* $\text{FOC}[+; \text{MOD}]$ *that defines Dyck-1.*

*Proof.* Suppose that such a sentence $\sigma$ exists. Chiang et al. (2023) show that there exists an $M$ such that $\sigma$ cannot distinguish between two strings that differ only by swapping symbols exactly $M$ positions apart. Since $(^M)^M \in$ Dyck-1 and $)(^{M-1}()^{M-1} \notin$ Dyck-1 but $\sigma$ cannot distinguish them, we have a contradiction. □

On the other hand, it does not seem that $\mathbf{K}_t[\#]$ can express Dyck-2, so a treatment of this language would be an informative target for future work. We note that Yao et al. (2021) shows transformers can recognize Dyck-2, but using hard attention.

Indeed, reconciling the $\mathbf{K}_t[\#]$ bound on softmax transformers with other bounds on hard-attention transformers poses challenges. For instance, $\mathbf{K}_t[\#]$ seems to be incomparable with **FO**$[<]$ (Angluin et al., 2023). $\mathbf{K}_t[\#]$ can express Dyck-1 but **FO**$[<]$ cannot, and **FO**$[<]$ can express $\Sigma^* aa \Sigma^*$, which $\mathbf{K}_t[\#]$ does not seem able to do.

Furthermore, we also note that $\mathbf{K}_t[\#]$ is a strict subset of LTL($\mathbf{C}$,+) which Barceló et al. (2024) show is a lower bound on average-hard attention transformer expressivity. However, average-hard attention transformers are not known to be either a subset or superset of standard, soft-attention transformers. More comments on hard and soft attention can be found in Section 2.1. Understanding the precise connection between AHATs and soft attention transformers is left for future exploration, and is expected to require modifications to the standard architecture in order to derive an exact inclusion.

As a final note, there appears to be a sizeable gap between the lower bound of $\mathbf{K}_t[\#]$ and the upper bound of **FOM**[BIT] shown in Merrill and Sabharwal (2023). In particular, the latter can express integer multiplication, but we suspect $\mathbf{K}_t[\#]$ cannot. Incidentally, transformer models are observed to have difficulty with integer multiplication (Dziri et al., 2024), so harmonizing these theoretical and empirical results would be helpful to paint a clearer picture of transformer expressivity.

## 7 From Fixed-Precision Transformers to $\mathbf{K}_t[\#]$

In practice, transformers use fixed-precision numbers. We adapt the proof that $\text{FOC}[+]$ can simulate fixed-precision soft attention transformers (Chiang et al., 2023) to show that $\mathbf{K}_t[\#]$ can simulate fixed-precision, masked soft attention transformers. If $\mathbf{K}_t[\#]$ is extended with modular predicates, it can also simulate sinusoidal positional encodings.

**Theorem 7.1.** $\mathbf{K}_t[\#]$ *can simulate fixed-precision masked soft attention transformers without positional encodings, and* $\mathbf{K}_t[\#; \text{MOD}]$ *can simulate fixed-precision masked soft attention transformers with sinusoidal positional encodings.*

*Proof.* See Appendix C. □

# 8 Autoregressive C-RASP Programs

With a simple extension, we can use **C-RASP** to construct decoder language models.

**Definition 8.1** (**C-RASP** Language Model). Assume a **C-RASP** program over alphabet $\Sigma$ has vectors $N_a(i)$ defined for each $a \in \Sigma \cup \{\text{EOS}\}$. Then a **C-RASP** program can be converted to an autoregressive language model with greedy decoding in the following manner

1. Translate the program into a transformer.
2. Append a linear layer, which selects the dimensions which simulate $N_a(i)$
3. Consider last position as a uniform probability distribution: dimensions with $N_a(|w|) = 1$ will have the same maximum probability score, and all those with $N_a(|w|) = 0$ will have the minimum probability.
4. Run the transformer on $w$, and then select an $a$ such that the $N_a$ dimension holds the maximum probability, with arbitrary tie-breaking. Append $a$ to $w$, and repeat until EOS is selected.

We say a **C-RASP** language model assigns nonzero probability $p$ to word $w = w_0 \cdots w_{n-1}$ iff $w, i \vDash N_{w_{i+1}}$ for all $0 \le i < n - 1$, and $w, n - 1 \vDash N_{\text{EOS}}$. We say a **C-RASP** Language Model recognizes language $L$ whenever it assigns nonzero probability to $w$ iff $w \in L$.

In logical terms, we can construct a **C-RASP** language model to recognize $L$ iff there exists a formula $\phi$ of $\mathbf{K}_t[\#]$ which recognizes $L$ and for all $a \in \Sigma$ we can define formulas $N_a$ such that $w \vDash N_a \iff \exists w'.waw' \vDash \phi$. In this case, $N_{\text{EOS}} = \phi$

**Example 8.2.** Append to the end of Dyck program in Example 3.2 the following operations:

$$N_((i) := \neg Q_{\text{EOS}}(i)$$
$$N_)(i) := \neg Q_{\text{EOS}}(i) \wedge C_)(i) < C_((i)$$
$$N_{\text{EOS}}(i) := D(i)$$

**Corollary 8.3.** *For every piecewise testable language L, there exists a **C-RASP** Language Model which recognizes L.*

*Proof.* This is an immediate consequence of Lemma 4.2. For any piecewise testable language $L$ we can define a formula $\phi$ of $\mathbf{K}_t[\#]$ which recognizes $L$. Next, observe that it is always the case that $\exists w'.waw' \vDash \phi$. Finally, define $N_{\text{EOS}} = \phi$ and $N_a = \top$ for all $a \in \Sigma$. $\square$

# 9 Concluding Remarks

We have introduced the temporal counting logic $\mathbf{K}_t[\#]$ alongside the RASP variant **C-RASP** and proved that they are the best-known lower bound on the expressivity of future-masked soft attention transformers, with unbounded input size. Unlike previous results, we have made minimal extra assumptions about transformers, so all formulas in $\mathbf{K}_t[\#]$ can compile directly into standard transformers. As such, an implementation of **C-RASP** should prove appropriate for constructing transformers to run experiments on, and further theoretical analysis of $\mathbf{K}_t[\#]$ and its extensions should shed light on the expressive power of transformers.

# 10 Acknowledgements

We would like to thank Dana Angluin, Pablo Barceló, Stephen Bothwell, Peter Cholak, Dakotah Lambert, Anthony Widjaja Lin, Anand Pillay, Jon Rawski, Darcey Riley, Ken Sible, Aarohi Srivastava, Lena Strobl, and Chihiro Taguchi for their feedback and support.

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

# A  Proofs Related to C-RASP

## A.1  Extensions

**Lemma A.1.** *Consider an extension of **C-RASP** in which the counting operation has the form*

$$C(i) := \#_2 \left[ j \leq i \right] \; F(i, j)$$

*which allows $F$ to be a Boolean combination of **C-RASP** operations $P(i)$ and $P(j)$ evaluated at both $i$ and $j$. Any program with this extended operation is actually equivalent to a **C-RASP** program with only the normal counting operation.*

*Proof.* Let $F$ be a Boolean combination of $P_1, \ldots, P_k$ evaluated at either $i$ or $j$. First, observe that $C(i) = C_1(i) + C_2(i) - C_3(i)$ where

$$
\begin{aligned}
C(i) &:= \#_2 \left[ j \leq i \right] \; F_1(i, j) \vee F_2(i, j) \\
C_1(i) &:= \#_2 \left[ j \leq i \right] \; F_1(i, j) \\
C_2(i) &:= \#_2 \left[ j \leq i \right] \; F_2(i, j) \\
C_3(i) &:= \#_2 \left[ j \leq i \right] \; F_1(i, j) \wedge F_2(i, j)
\end{aligned}
$$

Now, write $F$ in DNF and split using the above. Now every single counting operation is of the form

$$C(i) := \#_2 \left[ j \leq i \right] \; \left( \bigwedge_{x \in I} P_x(i) \wedge \bigwedge_{x \in J} P_x(j) \right)$$

Where $I$ and $J$ store the indices of **C-RASP** operations which depend on $i$ and $j$, respectively, within the counting operation. Observe then that this is equivalent to

$$C(i) := \# \left[ j \leq i \right] \; \left( \bigwedge_{x \in I \cap J} P_x(j) \right)$$

Thus, every $\#_2$ operation can be factored out as a sequence of normal $\#$ operations.  $\square$

## A.2  More C-RASP Examples

We sometimes put multiple **C-RASP** operations on the same line for brevity.

*a\*b\* over $\Sigma = \{a, b\}$*

$$
\begin{aligned}
C_a(i) &:= \# \left[ j \leq i \right] \; Q_a(j) && \text{\# positions with } a\text{'s} \\
C_b(i) &:= \# \left[ j \leq i \right] \; Q_b(j) && \text{\# positions with } b\text{'s} \\
V(i) &:= Q_a(i) \wedge C_b(i) \geq 1 && \textit{Violation}: \text{an } a \text{ has } b\text{'s preceding it} \\
C_V(i) &:= \# \left[ j \leq i \right] \; V(j) && \text{\# } \textit{Violations} \\
Y(i) &:= C_V(i) = 0 && \text{Zero } \textit{Violations}
\end{aligned}
$$

$a^*b^*a^*$ over $\Sigma = \{a, b\}$

$$
\begin{aligned}
C_a(i) &:= \#[j \le i]\ Q_a(j) && \text{\# positions with } a\text{'s} \\
C_b(i) &:= \#[j \le i]\ Q_b(j) && \text{\# positions with } b\text{'s} \\
BA(i) &:= Q_a(i) \wedge C_b(i) \ge 1 && \text{A subsequence } ba \text{ ends at } i \\
C_{ba}(i) &:= \#[j \le i]\ BA(j) && \text{\# ends of subsequence } ba \\
BAB(i) &:= Q_b(i) \wedge C_{ba} \ge 1 && \text{the subsequence } bab \text{ ends at } i \\
C_{bab}(i) &:= \#[j \le i]\ BAB(j) && \text{\# ends of subsequence } bab \\
Y(i) &:= C_{bab}(i) = 0 && \text{There are no subsequences } bab
\end{aligned}
$$

$a^n b^n c^n$ over $\Sigma = \{a, b, c\}$

$$
\begin{aligned}
C_a(i) &:= \#[j \le i]\ Q_a(j) && \text{\# positions with } a\text{'s} \\
C_b(i) &:= \#[j \le i]\ Q_b(j) && \text{\# positions with } b\text{'s} \\
C_c(i) &:= \#[j \le i]\ Q_c(j) && \text{\# positions with } c\text{'s} \\
A(i) &:= C_b(i) + C_c(i) = 0 && \text{No preceding } b\text{'s or } c\text{'s} \\
B(i) &:= C_c(i) = 0 && \text{No preceding } c\text{'s} \\
C_A(i) &:= \#[j \le i]\ Q_a(j) \wedge A(j) && \text{\# } a\text{'s with no preceding } b\text{'s or } c\text{'s} \\
C_B(i) &:= \#[j \le i]\ Q_b(j) \wedge B(j) && \text{\# } b\text{'s with no preceding } c\text{'s} \\
G_a(i) &:= C_A(i) = C_a(i) && \text{no } a\text{'s have preceding } b\text{'s or } c\text{'s} \\
G_b(i) &:= C_B(i) = C_b(i) && \text{no } b\text{'s have preceding } c\text{'s} \\
G_{abc}(i) &:= C_a(i) = C_b(i) = C_c(i) && \text{same number of } a\text{'s, } b\text{'s, } c\text{'s} \\
Y(i) &:= G_a(i) \wedge G_b(i) \wedge G_{abc}(i) && \text{Correct order and number of symbols}
\end{aligned}
$$

$hello$ over $\Sigma = \{e, h, l, o\}$

$$
\begin{aligned}
C_e(i) &:= \#[j \le i]\ Q_e(j) && \text{\# positions with } e\text{'s} \\
C_h(i) &:= \#[j \le i]\ Q_h(j) && \text{\# positions with } h\text{'s} \\
C_l(i) &:= \#[j \le i]\ Q_l(j) && \text{\# positions with } l\text{'s} \\
C_o(i) &:= \#[j \le i]\ Q_o(j) && \text{\# positions with } o\text{'s} \\
C_\Sigma(i) &:= \#[j \le i]\ 1 && \text{\# symbols in string} \\
HE(i) &:= Q_e(i) \wedge C_h(i) = 1 && \text{A subsequence } he \text{ ends at } i \\
C_{he}(i) &:= \#[j \le i]\ HE(j) && \text{\# ends of subsequence } he \\
HEL(i) &:= Q_l(i) \wedge C_{he}(i) = 1 && \text{A subsequence } hel \text{ ends at } i \\
C_{hel}(i) &:= \#[j \le i]\ HEL(j) && \text{\# ends of subsequence } hel \\
HELLO(i) &:= Q_o(i) \wedge C_{hel}(i) = 2 && \text{A subsequence } hello \text{ ends at } i \\
Y(i) &:= HELLO(i) \wedge C_\Sigma(i) = 5 && \text{Length 5 and contains subsequence } hello
\end{aligned}
$$

As a potential point of clarification for the $HELLO(i)$ line, observe that if a string contains two positions that are the end of a subsequence $hel$, then that string must contain the subsequence $hell$.

### A.3 Equivalence with $\mathbf{K}_t[\#]$

**Theorem 4.3.** A **C-RASP** program recognizes language $L$ iff a $\mathbf{K}_t[\#]$ formula defines $L$. More precisely, given alphabet $\Sigma$, for any $\mathbf{K}_t[\#]$ formula $\phi$ there is a **C-RASP** program $P$ such that $w \in \Sigma^*$ end-satisfies $\phi$ iff $w$ is accepted by $P$, and vice versa.

*Proof.* It is straightforward to translate $\mathbf{K}_t[\#]$ formulas into **C-RASP** programs.

In the other direction, we induct on the length of **C-RASP** program $\mathcal{P} = P_1, \ldots, P_n$. Assume that Boolean operations $P_k(i)$ are simulated by formulas $\hat{P}_k$, and count operations $C_k(i) = \#\,[j \leq i]\ V(j)$ are simulated by terms $\hat{C}_k$. This is straightforward, but there are many cases. The main idea is that whenever we have a conditional or min/max operation, we divide the formula into two cases depending on what the result of the operation should be.

- If $P_{k+1}$ is a count-valued vector, it is not used for string acceptance as defined in 3.1. Thus the formula for this is the formula for the last Boolean-valued vector, by the IH.

- If $P_{k+1}(i) = Q_a(i)$, let $\hat{P}_{k+1} = Q_a$.

- If $P_{k+1}(i) = \neg P_\ell(i)$ let $\hat{P}_{k+1} = \neg \hat{P}_\ell$.

- If $P_{k+1}(i) = P_\ell(i) \wedge P_m(i)$ let $\hat{P}_{k+1} = \hat{P}_\ell \wedge \hat{P}_m$.

- If $P_{k+1}(i) = C_\ell(i) \leq C_m(i)$ we need to divide on cases to handle if $C_\ell(i)$ is a conditional or min/max, as $\mathbf{K}_t[\#]$ does not have these terms built in. This is straightforward, but there are many cases.

  - If $C_\ell(i) = \#\,[j \leq i]\ P_1(i)$, let $\hat{C}_\ell = \#\big[\hat{P}_1\big]$. Then:
    - If $C_m(i) = \#\,[j \leq i]\ P_2(i)$, let $\hat{P}_{k+1} = \hat{C}_\ell \leq \#\big[\hat{P}_2\big]$.
    - If $C_m(i) = P_2(i)\ ?\ C_3(i) : C_4(i)$, let $\hat{P}_{k+1} = (\hat{P}_2 \wedge \hat{C}_\ell \leq \hat{C}_3) \vee (\neg \hat{P} \wedge \hat{C}_\ell \leq \hat{C}_4)$.
    - If $C_m(i) = \min(C_3(i), C_4(i))$, let $\hat{P}_{k+1} = (\hat{C}_3 \leq \hat{C}_4 \wedge \hat{C}_\ell \leq \hat{C}_3) \vee (\hat{C}_4 \leq \hat{C}_3 \wedge \hat{C}_\ell \leq \hat{C}_4)$.
    - If $C_m(i) = \max(C_3(i), C_4(i))$, let $\hat{P}_{k+1} = (\hat{C}_3 \geq \hat{C}_4 \wedge \hat{C}_\ell \leq \hat{C}_3) \vee (\hat{C}_4 \geq \hat{C}_3 \wedge \hat{C}_\ell \leq \hat{C}_4)$.
    - If $C_m(i) = c$, for $c \in \mathbb{N}$ let $\hat{P}_{k+1} = \hat{C}_\ell \leq c$.
  - If $C_\ell(i) = P_1(i)\ ?\ C_1(i) : C_2(i)$, then:
    - If $C_m(i) = \#\,[j \leq i]\ P_2(i)$, let $\hat{P}_{k+1} = (\hat{P}_1 \wedge \hat{C}_1 \leq \#\big[\hat{P}_2\big]) \vee (\neg \hat{P}_1 \wedge \hat{C}_2 \leq \#\big[\hat{P}_2\big])$.
    - If $C_m(i) = P(i)\ ?\ C_3(i) : C_4(i)$, let $\hat{P}_{k+1} = (\hat{P}_1 \wedge \hat{P}_2 \wedge \hat{C}_1 \leq \hat{C}_3) \vee (\hat{P}_1 \wedge \neg \hat{P}_2 \wedge \hat{C}_1 \leq \hat{C}_4) \vee (\neg \hat{P}_1 \wedge \hat{P}_2 \wedge \hat{C}_2 \leq \hat{C}_3) \vee (\neg \hat{P}_1 \wedge \neg \hat{P}_2 \wedge \hat{C}_2 \leq \hat{C}_4)$.
    - If $C_m(i) = \min(C_3(i), C_4(i))$, let $\hat{P}_{k+1} = (\hat{P}_1 \wedge \hat{C}_3 \leq \hat{C}_4 \wedge \hat{C}_1 \leq \hat{C}_3) \vee (\hat{P}_1 \wedge \hat{C}_3 \geq \hat{C}_4 \wedge \hat{C}_1 \leq \hat{C}_4) \vee (\neg \hat{P}_1 \wedge \hat{C}_3 \leq \hat{C}_4 \wedge \hat{C}_2 \leq \hat{C}_3) \vee (\neg \hat{P}_1 \wedge \hat{C}_3 \geq \hat{C}_4 \wedge \hat{C}_2 \leq \hat{C}_4)$.
    - If $C_m(i) = \max(C_3(i), C_4(i))$, let $\hat{P}_{k+1} = (\hat{P}_1 \wedge \hat{C}_3 \geq \hat{C}_4 \wedge \hat{C}_1 \leq \hat{C}_3) \vee (\hat{P}_1 \wedge \hat{C}_3 \leq \hat{C}_4 \wedge \hat{C}_1 \leq \hat{C}_4) \vee (\neg \hat{P}_1 \wedge \hat{C}_3 \geq \hat{C}_4 \wedge \hat{C}_2 \leq \hat{C}_3) \vee (\neg \hat{P}_1 \wedge \hat{C}_3 \leq \hat{C}_4 \wedge \hat{C}_2 \leq \hat{C}_4)$.
    - If $C_m(i) = c$, for $c \in \mathbb{N}$ let $\hat{P}_{k+1} = (\hat{P}_1 \wedge \hat{C}_1 \leq c) \vee (\neg \hat{P}_1 \wedge \hat{C}_2 \leq c)$.
  - If $C_\ell(i) = \min(C_1(i), C_2(i))$, then:
    - If $C_m(i) = \#\,[j \leq i]\ P_2(i)$, let $\hat{P}_{k+1} = (\hat{C}_1 \leq \hat{C}_2 \wedge \hat{C}_1 \leq \#\big[\hat{P}_2\big]) \vee (\hat{C}_1 \geq \hat{C}_2 \wedge \hat{C}_2 \leq \#\big[\hat{P}_2\big])$.
    - If $C_m(i) = P(i)\ ?\ C_3(i) : C_4(i)$, let $\hat{P}_{k+1} = (\hat{C}_1 \leq \hat{C}_2 \wedge \hat{P}_2 \wedge \hat{C}_1 \leq \hat{C}_3) \vee (\hat{C}_1 \leq \hat{C}_2 \wedge \neg \hat{P}_2 \wedge \hat{C}_1 \leq \hat{C}_4) \vee (\hat{C}_1 \geq \hat{C}_2 \wedge \hat{P}_2 \wedge \hat{C}_2 \leq \hat{C}_3) \vee (\hat{C}_1 \geq \hat{C}_2 \wedge \neg \hat{P}_2 \wedge \hat{C}_2 \leq \hat{C}_4)$.
    - If $C_m(i) = \min(C_3(i), C_4(i))$, let $\hat{P}_{k+1} = (\hat{C}_1 \leq \hat{C}_2 \wedge \hat{C}_3 \leq \hat{C}_4 \wedge \hat{C}_1 \leq \hat{C}_3) \vee (\hat{C}_1 \leq \hat{C}_2 \wedge \hat{C}_3 \geq \hat{C}_4 \wedge \hat{C}_1 \leq \hat{C}_4) \vee (\hat{C}_1 \geq \hat{C}_2 \wedge \hat{C}_3 \leq \hat{C}_4 \wedge \hat{C}_2 \leq \hat{C}_3) \vee (\hat{C}_1 \geq \hat{C}_2 \wedge \hat{C}_3 \geq \hat{C}_4 \wedge \hat{C}_2 \leq \hat{C}_4)$.
    - If $C_m(i) = \max(C_3(i), C_4(i))$, let $\hat{P}_{k+1} = (\hat{C}_1 \leq \hat{C}_2 \wedge \hat{C}_3 \geq \hat{C}_4 \wedge \hat{C}_1 \leq \hat{C}_3) \vee (\hat{C}_1 \leq \hat{C}_2 \wedge \hat{C}_3 \leq \hat{C}_4 \wedge \hat{C}_1 \leq \hat{C}_4) \vee (\hat{C}_1 \geq \hat{C}_2 \wedge \hat{C}_3 \geq \hat{C}_4 \wedge \hat{C}_2 \leq \hat{C}_3) \vee (\hat{C}_1 \geq \hat{C}_2 \wedge \hat{C}_3 \leq \hat{C}_4 \wedge \hat{C}_2 \leq \hat{C}_4)$.
    - If $C_m(i) = c$, for $c \in \mathbb{N}$ let $\hat{P}_{k+1} = (\hat{C}_1 \leq \hat{C}_2 \wedge \hat{C}_1 \leq c) \vee (\hat{C}_1 \geq \hat{C}_2 \wedge \hat{C}_2 \leq c)$.

- If $C_\ell(i) = \max(C_1(i), C_2(i))$, this identical to the previous case but just switch $\hat{C}_1 \leq \hat{C}_2$ with $\hat{C}_1 \geq \hat{C}_2$.
- If $C_\ell(i) = c$ for $c \in \mathbb{N}$, this should be straightforward given the above cases written out.

$\square$

**Lemma 6.1.** $\mathbf{K}_t[\#]$ contains FOC$[+]$, and $\mathbf{K}_t[\#; \text{MOD}]$ contains FOC$[+; \text{MOD}]$. In general, for any set $\mathcal{P}$ of unary predicates, the extension FOC$[+; \mathcal{P}]$ is contained in $\mathbf{K}_t[\#; \mathcal{P}]$, when defined as expected.

*Proof.* All FOC$[+]$ sentences can be rewritten in the following normal form (Chiang et al., 2023, Theorem 1):

$$\exists x_1 \ldots \exists x_n. \left( \bigwedge_i \exists^{=x_i} p.\psi_i(p) \wedge \chi(x_1, \ldots, x_n) \right)$$

where all $\psi_i$ and $\chi$ are quantifier-free, and $\chi$ is a set of linear constraints on $x_1 \ldots x_n$.

With respect to end-satisfiablity, this is simply equivalent to the $\mathbf{K}_t[\#]$ formula.

$$\chi\left(\#[\psi_1], \ldots, \#[\psi_n]\right)$$

The same is true for FOC$[+; \text{MOD}]$, if we extend $\mathbf{K}_t[\#]$ with modular predicates to $\mathbf{K}_t[\#; \text{MOD}]$ where $(w, i) \vDash \text{MOD}_m^k \iff i \equiv k \mod m$.

In general, from this normal form we can see that for any set $\mathcal{P}$ of unary predicates, the extension FOC$[+; \mathcal{P}]$ is contained in $\mathbf{K}_t[\#; \mathcal{P}]$, when defined appropriately. Furthermore, all of these will be $\mathbf{K}_t[\#]$ formulas of modal depth 1. $\square$

# B Proofs: From $\mathbf{K}_t[\#]$ to Transformers

In all proofs, we omit the scaling factor applied by LayerNorm, because it can be confusing and also does not affect the end result. However, whenever relevant, we mention how to account for this scaling factor.

Here, we recall the definition of LayerNorm

**Definition B.1.** LayerNorm is a function $f : \mathbb{R}^d \to \mathbb{R}^d$ such that

$$f(x) = \frac{x - \mu}{\sigma} \quad \text{where} \quad \mu = \frac{1}{d}\sum_{i=1}^{d} x_i \qquad \sigma = \sqrt{\frac{1}{d}\sum_{i=1}^{d}(x_i - \mu)^2}$$

Observe that if $\mu = 0$, LayerNorm only applies a scaling factor to all values in a position. Observe furthermore if the absolute value of all values in a position are equal, LayerNorm scales all of them to be $\pm 1$, which is important in Appendix B.3.

An essential part of our construction is access to a Boolean vector that is True at the BOS position and False everywhere else. We call this a *Start-Separating Vector*, adapting terminology from Merrill and Sabharwal (2024). Using the word embedding for BOS, we can construct a dimension that holds such a vector (ensuring $WE(w_i)$ is true at dimensions $2k_s - 1, 2k_s$ only when $w_i = \text{BOS}$).

$$
\begin{array}{c c}
& \begin{array}{c c c c}
1 & 2 & & n'
\end{array} \\
\begin{array}{c}
2k_s - 1 \\
2k_s
\end{array} &
\left[
\begin{array}{c c c c}
\vdots & \vdots & & \vdots \\
-1 & +1 & \cdots & +1 \\
+1 & -1 & \cdots & -1 \\
\vdots & \vdots & & \vdots
\end{array}
\right]
\end{array}
$$

### B.1 BOS **Handling Lemma**

**Lemma 5.3.** Using the word embedding and a single feed-forward layer, we can set the BOS position to False, without disturbing the Boolean value at any other position.

*Proof.* Construct a feed-forward layer which computes the min of every value in an odd dimension with the value in dimension $2k_s - 1$ of the Start-Separating Vector and the max of every value in an even dimension with the value in $2k_s$, as described in Lemma 5.5. Since we've maintained that all values are $\pm 1$ in our Boolean representation, this sets the BOS position to be False, while the others are unmodified. $\square$

### B.2 Counting Lemma

**Lemma 5.4.** Let $A_{2k-1:2k}, *$ store a Boolean vector as defined above. For any $i$, let $C_{k,i}$ be the number of positions $j \leq i$ such that $A_{2k-1:2k,j}$ is True. Then there is a transformer block that computes, at each position $i$, and in two other dimensions $2k' - 1, 2k'$, the values $-\frac{C_{k,i}}{i+1}$ and $\frac{C_{k,i}}{i+1}$.

*Proof.* First, recall the definition of a masked self attention layer $SA \colon \mathbb{R}^{d \times n} \to \mathbb{R}^{d \times n}$:

$$
SA(A) = \begin{bmatrix} c_o & \cdots & c_n \end{bmatrix} \text{ where}
$$
$$
W^{(Q)} \colon \mathbb{R}^d \to \mathbb{R}^{d_k}
$$
$$
W^{(K)} \colon \mathbb{R}^d \to \mathbb{R}^{d_k}
$$
$$
W^{(V)} \colon \mathbb{R}^d \to \mathbb{R}^d
$$
$$
s_{ij} = \frac{W^{(Q)} A_{*,i} \cdot W^{(K)} A_{*,j}}{\sqrt{d}}
$$
$$
c_i = \frac{\sum_{j=0}^{i} \exp(s_{ij}) W^{(V)} A_{*,j}}{\sum_{j=0}^{i} \exp(s_{ij})}
$$

To simulate a count term $\#[\phi] = C(i)$ of $\mathbf{K}_t[\#]$, we need to construct a transformer block such that if the Boolean values $\phi(i)$ are stored in some dimension $2k - 1, k$, we can compute $\frac{C(i)}{i+1}$ in some other dimensions $2k' - 1, k'$.

To achieve this, we only need *uniform attention* – that is, at each position $i$, we set $W^{(Q)} = W^{(K)} = \mathbf{0}$, which makes $s_{ij} = 0$ for all $i, j$. This spreads attention weight evenly across all positions $j \leq i$.

Furthermore, by setting $W^{(V)}$ as follows, we can add the value from position $i$ in dimensions $2k - 1, 2k$ to position $i$ in dimensions $2k' - 1, 2k'$.

$$W^{(V)} = \begin{array}{c} \\ \\ \\ \\ 2k-1 \\ 2k \\ \\ 2k'-1 \\ 2k' \\ \\ \end{array} \begin{array}{c} \overset{2k-1}{} \quad \overset{2k}{} \qquad \overset{2k'-1}{} \quad \overset{2k'}{} \\ \left[ \begin{array}{cccccc} \vdots & \vdots & & \vdots & \vdots & \\ \cdots & 0 & 0 & \cdots & 0 & 0 & \cdots \\ \cdots & 0 & 0 & \cdots & 0 & 0 & \cdots \\ & \vdots & \vdots & & \vdots & \vdots & \\ \cdots & 1 & 1 & \cdots & 0 & 0 & \cdots \\ \cdots & 1 & 1 & \cdots & 0 & 0 & \cdots \\ & \vdots & \vdots & & \vdots & \vdots & \end{array} \right] \end{array}$$

As such, we get the attention layer to compute the average of all unmasked positions $j \leq i$. Recalling the definition once more, the general expression reads:

$$c_{i,k} = \frac{\sum_{j=0}^{i} \exp(s_{ij})[W^{(V)} A_{*,j}]_k}{\sum_{j=0}^{i} \exp(s_{ij})}$$

Then, with the use of uniform attention and our carefully constructed $W^{(V)}$, we compute in position $i$ of dimension $k$, the value $c_{i,k}$, which is the average of all values up to position $i$ in dimension $k$, The expression reduces to:

$$c_{i,k} = \frac{\sum_{j=0}^{i} \exp(s_{ij})[W^{(V)} A_{*,j}]_k}{\sum_{j=0}^{i} \exp(s_{ij})} = \frac{\sum_{j=0}^{i}[W^{(V)} A_{*,j}]_k}{\sum_{j=0}^{i} 1} = \frac{\sum_{j=0}^{i} A_{k,j}}{\sum_{j=0}^{i} 1} = \frac{\sum_{j=0}^{i} A_{k,j}}{i+1}$$

Before moving on, note that we represent Booleans as $-1, +1$ instead of $0, 1$, and we'll also write the sum of positions $j \leq i$ that hold a True as $C_{i,k}$. Here we write out the resulting tensor after the described self-attention layer showing the relevant dimensions at positions $0, 1, \ldots, n'$. We write $B_i \in \{0,1\}$ be the Boolean value at position $i$.

$$\begin{array}{c} \\ 2k-1 \\ 2k \\ \\ 2k'-1 \\ 2k' \\ \\ \end{array} \begin{array}{c} \overset{1}{} \qquad \overset{2}{} \qquad\qquad \overset{n'}{} \\ \left[ \begin{array}{cccc} \vdots & \vdots & & \vdots \\ +1 & -2B_1+1 & \cdots & -2B_n+1 \\ -1 & +2B_1-1 & \cdots & +2B_n-1 \\ \vdots & \vdots & & \vdots \\ 0 & -2\frac{C_{1,d'}}{2}+1 & \cdots & -2\frac{C_{n,d'}}{n+1}+1 \\ 0 & +2\frac{C_{1,d'}}{2}-1 & \cdots & +2\frac{C_{n,d'}}{n+1}-1 \\ \vdots & \vdots & & \vdots \end{array} \right] \end{array}$$

Note, however, that instead of the desired value $\frac{C_{i,k}}{i+1}$, we have actually computed $2\frac{C_{i,k}}{i+1} - 1$. We use a feed-forward layer to undo this transformation by subtracting (or adding) 1 and then dividing by 2 in dimension $d$ (or $d+1$). Then, it is straightforward to construct a feed-forward layer that simply adds dimension $2k_0 - 1$ to $2k - 1$ and $2k_0$ to $2k$ in order to remove the $\pm$, and then divides by 2 to get the result:

$$
\begin{array}{c}
\phantom{2k'-1} \\
2k'-1 \\
2k' \\
\phantom{2k'} \\
\end{array}
\begin{array}{cccc}
1 & 2 & & n' \\
\begin{bmatrix}
\vdots & \vdots & & \vdots \\
0 & -\frac{C_{1,d'}}{2} & \cdots & -\frac{C_{n,d'}}{n+1} \\
0 & +\frac{C_{1,d'}}{2} & \cdots & +\frac{C_{n,d'}}{n+1} \\
\vdots & \vdots & & \vdots
\end{bmatrix}
\end{array}
$$

As a final note, which is relevant for the later subsection on arithmetic comparison, we will make every self-attention layer compute a counting operation over the start-separating vector (Appendix B.1) in order to compute the value $\frac{1}{i+1}$ at every position $i$ in some dimension. □

### B.3 Feed-Forward Lemma

**Lemma 5.5.** The following position-wise operations can be simulated by a single transformer block, using existing dimensions as input and a fresh dimension as output: addition $(+)$, subtraction $(-)$, comparison $(\leq)$, and Boolean operations $(\wedge, \neg)$. Arbitrary Boolean expressions in DNF can be simulated using two blocks.

*Proof.* A two-layer feed-forward network with ReLU activations on the first layer and none on the second layer can compute any continuous piecewise linear function with a finite number of pieces, or CPWL (Arora et al., 2018). All of these operations are CPWL, but we give explicit constructions for concreteness, and the case of comparisons requires special care.

We consider each type of operation in turn. All constructions only use the feed-forward layer. We can force the self-attention layer to perform no changes to the residual stream by setting all weights to 0; the residual connection then adds the original values back in.

*Addition and subtraction:* These operations are straightforward: put 1's in $W_1$ such that we add the appropriate values to the fresh dimension, and negate using $W_2$ if needed.

*Min/Max:* Recall that $\mathrm{ReLU}(x) = \max(0, x)$. Then $\min(x, y) = x - \mathrm{ReLU}(x - y)$.

– If $x < y$, then $\mathrm{ReLU}(x - y) = 0$, so $x - \mathrm{ReLU}(x - y) = x = \min(x, y)$.
– If $x \geq y$, then $\mathrm{ReLU}(x - y) = x - y$, so $x - \mathrm{ReLU}(x - y) = x - (x - y) = y = \min(x, y)$.

Similarly, $\max(x, y) = x + \mathrm{ReLU}(y - x)$.

Therefore there exist FFNs to compute the min or max of two real numbers:

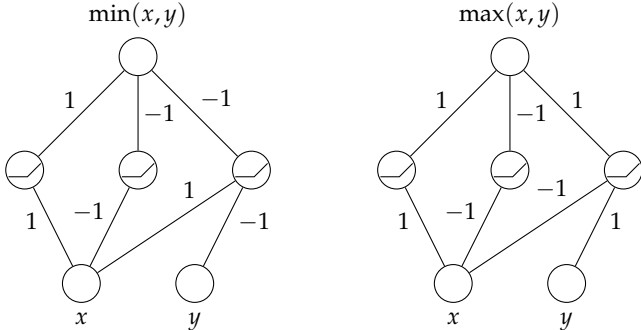

To compute the min of two count terms, we compute the min of their positive components and the max of their negative components. Similarly for the max of two count terms.

*Comparison:* This requires access to $\pm\frac{1}{i+1}$ in some dimensions $2k_0 - 1, 2k_0$. This is easily achieved by requiring every self-attention layer to perform a counting operation over the start-separating vector (Appendix B.1).

First we explain how to simulate a comparison of two count terms $C_1(i) \leq C_2(i)$. Then, we describe how to extend this to compare linear combinations of count terms.

Suppose that we want to compare $C_1$ and $C_2$ in dimensions $2k_1 - 1, 2k_1$ and $2k_2 - 1, 2k_2$, and put the result in dimension $2k_3 - 1, 2k_3$. Initially, the residual stream looks like this:

$$
\begin{array}{r}
\\
2k_0 - 1 \\
2k_0 \\
\\
2k_1 - 1 \\
2k_1 \\
\\
2k_2 - 1 \\
2k_2 \\
\\
2k_3 - 1 \\
2k_3 \\
\\
\end{array}
\begin{bmatrix}
& \vdots & \\
\cdots & -\frac{1}{i+1} & \cdots \\
\cdots & +\frac{1}{i+1} & \cdots \\
& \vdots & \\
\cdots & -\frac{C_1(i)}{i+1} & \cdots \\
\cdots & +\frac{C_1(i)}{i+1} & \cdots \\
& \vdots & \\
\cdots & -\frac{C_2(i)}{i+1} & \cdots \\
\cdots & +\frac{C_2(i)}{i+1} & \cdots \\
& \vdots & \\
\cdots & 0 & \cdots \\
\cdots & 0 & \cdots \\
& \vdots & \\
\end{bmatrix}
\begin{array}{c} i \\ \end{array}
$$

We construct a feed-forward layer that computes the function:

$$
\text{gtz}(X(i)) = \min\left(\frac{0.5}{i+1}, \frac{X(i)}{i+1} - \frac{0.5}{i+1}\right) - \min\left(0, \frac{X(i)}{i+1}\right).
$$

Observe that $\text{gtz}(C_2(i) - C_1(i) + 0.5)$ equals $\frac{0.5}{i+1}$ if $C_1(i) \leq C_2(i)$, and $-\frac{0.5}{i+1}$ otherwise. This is because the counts $C_1(i), C_2(i)$ must be integers, so if $C_1(i) \leq C_2(i)$, then $C_2(i) - C_1(i) + 0.5 \geq 0.5$, and the expression will evaluate to $\frac{0.5}{i+1}$. Otherwise, $C_2(i) - C_1(i) + 0.5 < -0.5$, and the expression will evaluate to $-\frac{0.5}{i+1}$.

It is straightforward, then, to use the construction for min/max from above to produce a feed-forward layer that computes $\text{gtz}(C_2(i) - C_1(i))$. Essentially, we use $W_1$ to compute the values (using the pre-existing values from the residual stream)

$$
\frac{0.5}{i+1}, \frac{C_2(i) - C_1(i) + 0.5}{i+1}, -\frac{C_2(i) - C_1(i) + 0.5}{i+1}
$$

Then we use $W_2$ to compute

$$\frac{0.5}{i+1} + \text{ReLU}\left(\frac{0.5}{i+1} - \frac{C_2(i) - C_1(i) + 0.5}{i+1}\right) - \text{ReLU}\left(\frac{0.5}{i+1} - \frac{C_2(i) - C_1(i) - 0.5}{i+1}\right)$$

which equals $\text{gtz}(C_2(i) - C_1(i))$ as desired.

Similarly, it is straightforward to construct a feed-forward layer to compare linear combinations of count terms. That is, for disjoint sets of indices $K_1$ and $K_2$, to compute

$$\text{gtz}\left(\sum_{k \in K_2} c_k \cdot C_k(i) - \sum_{k \in K_1} c_k \cdot C_k(i)\right).$$

So we can construct a feed-forward layer $f : \mathbb{R}^d \to \mathbb{R}^d$ that computes in each dimension $i$ the following

$$f\left(\begin{bmatrix} v_1 \\ v_2 \\ \vdots \\ 0 \\ 0 \\ \vdots \\ v_{d-1} \\ v_d \end{bmatrix}\right) = \begin{bmatrix} \text{gtz}(v_1) \\ \text{gtz}(v_2) \\ \vdots \\ \text{gtz}\left(\sum_{k \in K_2} c_k \cdot C_k(i) - \sum_{k \in K_1} c_k \cdot C_k(i)\right) \\ \text{gtz}\left(\sum_{k \in K_2} c_k \cdot C_k(i) - \sum_{k \in K_1} c_k \cdot C_k(i)\right) \\ \vdots \\ \text{gtz}(v_{d-1}) \\ \text{gtz}(v_d) \end{bmatrix}.$$

This truncates all positive values in the residual stream at this point to be $\frac{0.5}{i+1}$ at position $i$, and all nonpositive values to be $-\frac{0.5}{i+1}$. As a result, the next application of LayerNorm (with appropriate parameter settings) scales every single value to $\pm 1$. In particular, all previously-computed Boolean values are preserved, and the newly-computed dimensions $2k_3 - 1, 2k$ hold the correct Boolean value based on the desired comparison

As a side effect, all previously-computed counts also get changed to $\pm 1$, but we do not need these counts any longer due to how we organized the construction in Theorem 5.7.

*Boolean operations:* The Boolean operations $\wedge$ and $\neg$ can be computed by FFNNs with ReLU activations. Conjunction ($\wedge$) is equivalent to min/max:

$$\begin{bmatrix} \vdots \\ -2B_1 + 1 \\ 2B_1 - 1 \\ \vdots \end{bmatrix} \wedge \begin{bmatrix} \vdots \\ -2B_2 + 1 \\ 2B_2 - 1 \\ \vdots \end{bmatrix} = \begin{bmatrix} \vdots \\ \max(-2B_1 + 1, -2B_2 + 1) \\ \min(\ 2B_1 - 1,\ \ 2B_2 - 1) \\ \vdots \end{bmatrix}.$$

Logical negation ($\neg$) is equivalent to arithmetic negation, or swapping the positive and negative components:

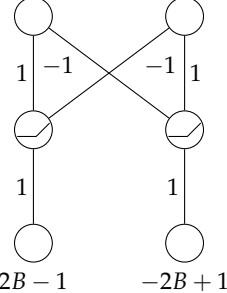

For an arbitrary Boolean formula, convert it to *canonical* disjunctive normal form, which is a disjunction $\phi_1 \vee \cdots \vee \phi_n$ of clauses, at most one of which can be true for any value of the inputs. Each clause is of the form $\phi_m = \bigwedge_{k \in K_m} B_k$, where each $B_k$ is an input or a negated input and $K_m$ is a set of indices for each clause. A slightly different construction can be used to compute $\wedge$ over inputs in $K_m$. Observe that:

$$\bigwedge_{k \in K_m} B_k = \text{ReLU}\left(\left(\sum_{k \in K_m} B_k\right) - (|K_m| - 1)\right)$$

And this can be computed for each clause using the first layer and ReLU of a feed-forward layer. Recall again that if a constant is fixed, we can retrieve it by multiplying the constant $\pm 1$ from dimensions $2k_0 - 1, 2k_0$ as described in Appendix B.2. Then, because at most one clause can be true, the sum of all clauses will either be 1 or 0. Then, we convert back to the $\pm 1$ representation of truth values.

$$\bigvee_{m=1}^{n}\left(\bigwedge_{k \in K_m} B_k\right) = 2 \cdot \left(\sum_{m=1}^{n} \text{ReLU}\left(\left(\sum_{k \in K_m} B_k\right) - (|K_m| - 1)\right)\right) - 1.$$

This can all be done in a single feed-forward layer.

$\square$

### B.4 Parallel Composition of Transformers

**Lemma B.2.** *Here we detail the claim in Theorem 5.7 that we can compose many transformers, into a larger transformer which simulates all the formulas the smaller transformers do, in parallel.*

*Two transformers $T_1$ and $T_2$ simulating formulas $\psi_1$ and $\psi_2$ (using the construction described in Theorem 5.7) can be composed into a single transformer $T_3$ that simulates both $\psi_1$ and $\psi_2$.*

*Proof.* We will just sketch out the construction. Let $T_1$ with $b_1$ blocks in $d_1$ dimensions and word embedding $\text{Emb}_1$ simulate $\phi_1$ and $T_2$ with $b_2$ blocks in $d_2$ dimensions and word embedding $\text{Emb}_2$ simulate $\phi_2$ in the manner described in Theorem 5.7. We can construct $T_1 \oplus T_2$ with $\max(b_1, b_2)$ blocks in $d_1 + d_2$ dimensions which simulates both $\phi_1$ and $\phi_2$, such that

$$(T_1 \oplus T_2)(w) = \begin{bmatrix} T_1(w) \\ T_2(w) \end{bmatrix}.$$

First, add layers that compute the identity function to the shallower transformer so that both have depth $\max(k_1, k_2)$.

Next, concatenate their word embedding vectors

$$(\text{Emb}_1 \oplus \text{Emb}_2)(a) = \begin{bmatrix} \text{Emb}_1(a) \\ \text{Emb}_2(a) \end{bmatrix}.$$

Then, at each level, we can compose the feed-forward networks $\text{FF}_1$ and $\text{FF}_2$ in parallel by creating a new FF with

$$W^{(1)} = \begin{bmatrix} W_1^{(1)} & \mathbf{0} \\ \mathbf{0} & W_2^{(1)} \end{bmatrix} \qquad\qquad b^{(1)} = \begin{bmatrix} b_1^{(1)} \\ b_2^{(1)} \end{bmatrix}$$

$$W^{(2)} = \begin{bmatrix} W_1^{(2)} & \mathbf{0} \\ \mathbf{0} & W_2^{(2)} \end{bmatrix} \qquad\qquad b^{(2)} = \begin{bmatrix} b_1^{(2)} \\ b_2^{(2)} \end{bmatrix}.$$

Furthermore, we can compose the self-attentions in exactly the same manner, as only uniform attention is used in the construction.

$$W^{(Q)} = \begin{bmatrix} W_1^{(Q)} & W_2^{(Q)} \end{bmatrix}$$

$$W^{(K)} = \begin{bmatrix} W_1^{(K)} & W_2^{(K)} \end{bmatrix}$$

$$W^{(Q)} = \begin{bmatrix} W_1^{(V)} & \mathbf{0} \\ \mathbf{0} & W_2^{(V)} \end{bmatrix}$$

It is straightforward to verify the correctness of this construction, given we follow the procedure in Theorem 5.7 organizing the simulation by modal depth. □

## C Fixed Precision Masked Transformers to $\mathbf{K}_t[\#; \mathsf{MOD}]$

**Definition C.1.** A *fixed-precision number* with $r$ integer bits and $s$ fractional bits is a number in $\mathbb{F}_{r,s} = \{i/2^s \mid -2^{r+s} \leq i < 2^{r+s}\}$. For any value $a \in \mathbb{F}_{r,s}$, we write $\langle a \rangle_b$ for the $b$-th bit of the two's complement representation of $a$. That is,

$$\langle a \rangle_b = \lfloor a \cdot 2^{-b} \rfloor - \lfloor a \cdot 2^{-b-1} \rfloor \cdot 2.$$

This is a two's complement representation.

It helps to access each individual bit of $x$.

**Proposition C.2.** *We write $x^b$ for the $b$-th bit of $x$, whenever $x$ is a fixed-precision number. Then, observe that we can write a formula $F_m(x) \iff x = F_m$ iff we can write formulas $F^b(x) \iff x^b = 1$.*

*Proof.* This should be clear by an example: 1001 in $\mathbb{F}_{5,0}$. We write $F_m(x) \iff F^0(x) \wedge \neg F^1(x) \wedge \neg F^2(x) \wedge F^3(x)$. □

Let us first define what it means for a formula of $\mathbf{K}_t[\#; \mathsf{MOD}]$ to simulate a Fixed Precision transformer.

**Definition C.3.** Let $T : \Sigma^n \to \mathbb{F}_{r,s}^{d \times n'}$ be a fixed-precision masked soft attention transformer defined exactly the same except we use $\mathbb{F}_{r,s}$ instead of $\mathbb{R}$. We say $T$ can be simulated in $\mathbf{K}_t[\#; \mathsf{MOD}]$ if for every $F_m \in \mathbb{F}_{r,s}$ and every dimension of $k$ of $T$ we can write a formula $\Phi_m^k$ such that

$$[T(\mathsf{BOS} \cdot w)]_{k,i+1} = F_m \iff w, i \vDash \Phi_m^k$$

Similarly, defining predicates $\beta_m^k(i)$ for the BOS position. Essentially this means that we can write a formula that tells us what value the transformer must output, given any input.

**Theorem 7.1.** $\mathbf{K}_t[\#]$ can simulate fixed-precision masked soft attention transformers without positional encodings, and $\mathbf{K}_t[\#; \mathsf{MOD}]$ can simulate fixed-precision masked soft attention transformers with sinusoidal positional encodings.

**Proposition C.4.** *Assume we have Boolean functions $F_m^k(i)$ which return true iff the value at position $i$ in dimension $k$ is $F_m \in \mathbb{F}_{r,s}$ Then any function of the form $f_1(x_1, \ldots, x_d) = f_2(x_1, \ldots, x_d)$, where the $x_k$ is the value at dimension $k$ in position $i$, can be written as a Boolean combination of $F_m^k(i)$.*

*Proof.* Essentially, this is $\mathbb{F}^n \times \mathbb{F}^n \to \{0, 1\}$, which only takes on finitely many values. Thus it is tedious, but straightforward, to enumerate all tuples of $x$ which should return true, and write a formula that returns the correct answer. Let

$$\mathcal{K} = \{(m_1, \ldots, m_d) \mid f_1(F_{m_1}, \ldots, F_{m_d}) = f_2(F_{m_1}, \ldots, F_{k=m_d})\}$$

That is, $\mathcal{K}$ stores all $n$-tuples $K$ of indices such that given an $n$-tuple of fixed-precision numbers which each have the corresponding index in $K$, the equality holds. Then write

$$\bigvee_{K \in \mathcal{K}} \left( \bigwedge_{k \leq d} F^k_{K_k}(i) \right)$$

This formula will return 1 iff $x_k$ in dimensions $k$ at position $i$ have the correct fixed-precision values in order to satisfy the equality. $\square$

As a result, for any function $f \colon \mathbb{F}^d_{r,s} \to \mathbb{F}^d_{r,s}$ we can write formulas $\phi(i)$ to check which fixed-precision value the output of the function is at position $i$, given the formulas that check the values of the inputs at position $i$. This means

**Lemma C.5.** *We can write formulas $FFN^k_m(i)$, for each feed-forward layer FFN to check whether the output at position $i$ in dimension $k$ is $F_m \in \mathbb{F}_{r,s}$. Same for LayerNorm $LN^k_m(i)$.*

*Proof.* This is a direct consequence of [Proposition C.4](#) because these functions all take a finite number of fixed-precision inputs, and have a finite number of outputs. $\square$

The same is not the case for self-attention layers, as we have no bound on the input length. However, count terms help us out here.

**Lemma C.6.** *Assume we are using $\mathbb{F}_{r,s}$ as our fixed-precision representation. Assume we have access to predicates $F^b(i)$ which tell us if the value at position $i$ has a 1 in the b-th bit of its fixed-precision representation. Then we can compute the following summation as a counting term*

$$2^s \cdot \sum_{j \leq i} X_j$$

*where $X_j$ is a value at position j.*

*Proof.* Observe this summation is equivalent to

$$\sum_{j \leq i} X_j = 2^0 \cdot \sum_{j \leq i} F^0(j) + 2^1 \cdot \sum_{j \leq i} F^1(j) \ldots + 2^{r+s-1} \cdot \sum_{j \leq i} F^{r+s-1}(j) - 2^{r+s} \cdot \sum_{j \leq i} F^{r+s}(j)$$

$$= \#[F^0] + 2\#[F^1] + \ldots + 2^{r+s-1}\#\left[F^{r+s-1}\right] - 2^{r+s}\#\left[F^{r+s}\right]. \qquad \square$$

As a clarifying note, recall we are using two's complement, which is why the most significant bit is subtracted.

**Lemma C.7.** *We can write formulas $C^k_m(i)$ which are true iff the output of a self-attention layer at dimension $k$ in position $i$ is $F_m \in \mathbb{F}_{r,s}$*

*Proof.* Recall the definition of a self-attention layer in a fixed-precision masked soft attention transformer, $SA \colon \mathbb{F}^{d \times n}_{r,s} \to \mathbb{F}^{d \times n}_{r,s}$:

$$SA(A) = [c_o \quad \cdots \quad c_n] \text{ where}$$
$$W^{(Q)} \colon \mathbb{F}^d_{r,s} \to \mathbb{F}^{d_k}_{r,s}$$
$$W^{(K)} \colon \mathbb{F}^d_{r,s} \to \mathbb{F}^{d_k}_{r,s}$$
$$W^{(V)} \colon \mathbb{F}^d_{r,s} \to \mathbb{F}^d_{r,s}$$
$$s_{ij} = \frac{W^{(Q)} A_{*,i} \cdot W^{(K)} A_{*,j}}{\sqrt{d}}$$
$$c_i = \frac{\sum_{j=0}^i \exp(s_{ij}) W^{(V)} A_{*,j}}{\sum_{j=0}^i \exp(s_{ij})}$$

More specifically, we want to define a formula $c_m^k(k)$ such that

$$c_m^k(i) \iff F_m \leq \frac{\sum_{j \leq i} e^{Q_i \cdot K_j} V_j^k}{\sum_{j \leq i} e^{Q_i \cdot K_j}} < F_m + 2^{-s}$$

where $V_j^k$ is the $k$-th component of the $W^{(V)} A_{*,j}$. The bounds are because division in $\mathbb{F}_{r,s}$ must perform some sort of rounding to the nearest number. We rearrange the equation to the following:

$$\phi_m^k(i) \iff \sum_{j \leq i} \left( F_m \cdot e^{Q_i \cdot K_j} \right) \leq \sum_{j \leq i} \left( e^{Q_i \cdot K_j} V_j^k \right) < \sum_{j \leq i} \left( (F_m + 2^{-s}) \cdot e^{Q_i \cdot K_j} \right).$$

Now observe that we can enumerate all the (finitely many) values that the $\mathbb{F}^d$ vector $Q_i$ as $(Q_x)_{x \leq (d(r+s))}$ in order to get an expression that only depends on $j$:

$$c_m^k(i) \iff \begin{cases} \left( \sum_{j \leq i} \left( F_m \cdot e^{Q_1 \cdot K_j} \right) \leq \sum_{j \leq i} \left( e^{Q_1 \cdot K_j} V_j^k \right) \right) & Q_i = Q_1 \\ \wedge \left( \sum_{j \leq i} \left( e^{Q_1 \cdot K_j} V_j^k \right) < \sum_{j \leq i} \left( (F_m + 2^{-s}) \cdot e^{Q_1 \cdot K_j} \right) \right) & \\[2ex] \sum_{j \leq i} \left( F_m \cdot e^{Q_2 \cdot K_j} \right) \leq \sum_{j \leq i} \left( e^{Q_2 \cdot K_j} V_j^k \right) & Q_i = Q_2 \\ \wedge \left( \sum_{j \leq i} \left( e^{Q_2 \cdot K_j} V_j^k \right) < \sum_{j \leq i} \left( (F_m + 2^{-s}) \cdot e^{Q_2 \cdot K_j} \right) \right) & \\ \quad \vdots \qquad\qquad\qquad \vdots \qquad\qquad\qquad \vdots & \\[1ex] \sum_{j \leq i} \left( F_m \cdot e^{Q_{d(r+s)} \cdot K_j} \right) \leq \sum_{j \leq i} \left( e^{Q_{d(r+s)} \cdot K_j} V_j^k \right) & Q_i = Q_{d(r+s)} \\ \wedge \left( \sum_{j \leq i} \left( e^{Q_{d(r+s)} \cdot K_j} V_j^k \right) < \sum_{j \leq i} \left( (F_m + 2^{-s}) \cdot e^{Q_{d(r+s)} \cdot K_j} \right) \right) & \end{cases}$$

This can straightforwardly be translated into a Boolean combination of formulas $c_{m,Q_x(i)}^k$, etc., which by Proposition C.4 are definable. Furthermore, by the same proposition it is also straightforward to construct a formula $QKV(j)$ that specifies the value of $e^{Q_x \cdot K_j} V_j^k$ at position $j$, as well as a formula $F_m QK(j)$ that specifies the value of $F_m \cdot e^{Q_{pd} \cdot K_j}$. This allows us to write the above expression in a form such that Lemma C.6 can be applied directly to both sides of the equation, thus computing both as count terms and allowing their comparison within $\mathbf{K}_t[\#; \text{MOD}]$. Notice that by applying Lemma C.6 we are implicitly scaling both sides up by $2^s$, but this is fine, as equality would still be preserved under this scaling factor.

$\square$

Finally, to complete the simulation it remains to show that we can write formulas that simulate the word embedding and positional encoding, which will be similar to the construction of Chiang et al. (2023).

**Lemma C.8.** *We can write $\mathbf{K}_t[\#]$ formulas $WE_m^k(i)$ that check that the $k$-th dimension of the embedding at position $i$ has the value $F_m \in \mathbb{F}_{r,s}$*

*Proof.* For $a \in \Sigma \cup \{\text{BOS}\}$, let $WE(a) = [F_{m_1,a}, \ldots, F_{m_d,a}]$ denote the fixed-precision word embedding for $a$. Then, using the same notation as earlier, we can write

$$WE_m^k(i) = \bigvee_{a \in \{a | WE(a)^k = F_m\}} Q_a(i).$$

Similarly, we define $\beta_m^k(i)$, which checks the word embedding for BOS has value $F_m$ at dimension $k$. Sinusoidal positional encodings can be described in the same manner, using modular predicates. □

Finally,

*Proof.* As a direct consequence of the above lemmas, we can write a formula of $\mathbf{K}_t[\#]$, which simulates a fixed-precision masked soft attention transformer without positional encodings. Adding modular predicates allows $\mathbf{K}_t[\#; \mathrm{MOD}]$ to simulate fixed-precision masked soft attention transformer with sinusoidal positional encodings. This formula will have modal depth $O(L)$ in the number of layers $L$ of the transformer, and width roughly $O((2^F)^d)$ in the precision $F$ and width $d$ of the transformer. □

