# OpenReview forum: "Counting Like Transformers: Compiling Temporal Counting Logic Into Softmax Transformers"
_colmweb.org/COLM/2024/Conference — COLM_

### Official Review · Reviewer_d2M3 · 2024-05-10

**Rating:** 8
**Confidence:** 4
**Ethics Flag:** 1

**Summary:**

This paper extends the original RASP language to a variant called C-RASP to express temporal counting logic $\textbf{K}_t$[#] and compile them to future-masked soft attention transformers with unbounded input sizes. They show that any constant precision softmax decoder Transformer can be expressed via the C-RASP language. Further, they show how to use $\textbf{C-RASP}$ to construct language models that via greedy decoding generate sentences that have properties specified in $\textbf{K}_t$[#].

**Questions To Authors:**

1. The paper mentions that any constant precision softmax decoder Transformer can be expressed via the C-RASP language. In the proof that compiles the C-RASP language, the transformer constructed has uniform attention. Does this mean it is possible to compile a general Transformer to a larger uniform attention Transformer with more precision? If such is the case, how large would the larger Transformer be if one would translate a 4-bit GPT-2 Transformer with 124M parameters as an example?

**Reasons To Accept:**

1. The paper makes a solid contribution to the softmax decoder Transformer's lower bound of expressivity. In particular, counting logic is required for a variety of tasks. RASP is known for its inability for seemingly simple addition, and the problem is addressed in this paper.

2. The paper is clear, with abundant examples. The proof in the appendix is also well organized.

**Reasons To Reject:**

1. The paper may require some rewriting to be understood by a broader audience. Currently, the reader may need a familiarity with representation theory and RASP language.

---

> ### Author Rebuttal · Authors · 2024-05-31
>
> Thank you very much for your positive review! We will try to make the final version of the paper more accessible to a broader CoLM audience.
>
> # Responses to questions:
>
> Yes, our results imply that any fixed-precision transformer can be converted to a uniform-attention transformer (albeit a very big one); in fact, this was already shown by [Chiang et al., 2023].
>
> To answer your interesting question, we can specify exactly where the parameters are distributed. The base GPT-2 model has 12 attention layers, 12 heads, and an embedding dimension size of 768, so we can give an idea on how many parameters a uniform-attention transformer needs to be to simulate 4-bit GPT-2 by estimating the number of layers, heads, and embedding dimensions it requires.
>
> Nearly all of the overhead comes from needing to preprocess the attention scores so they can be simulated with uniform attention. Essentially, we have to enumerate all $(2^4)^{768}$ possible query vectors, and write predicates to access each bit of each dimension of each vector. Therefore, as a ballpark estimate, $16^{768}\cdot4\cdot768$ embedding dimensions get added (this enumeration allows us to simulate all heads with one).
>
> Then, following Theorem 5.7, an upper bound of $4$ blocks are needed to simulate each fixed-precision attention layer. Altogether, we roughly estimate that it would take a transformer with embedding dimension of $16^{768}\cdot3072$, $48$ single-head attention layers (with query and key weights all zero), and $48$ FFN's with hidden dimension $16^{768}\cdot3072$ in order to simulate 4-bit GPT-2.
>
> [Chiang et al., 2023]: https://arxiv.org/abs/2301.10743

---

> > ### Author Response · Authors · 2024-06-07
> > **Thank you for your response**
> >
> > We thank the reviewer for the comments. As the discussion period is close to its end, we believe that we have addressed all of your questions, but please let us know if there are any remaining concerns.

---

### Official Review · Reviewer_QRfH · 2024-05-10

**Rating:** 7
**Confidence:** 3
**Ethics Flag:** 1

**Summary:**

This paper proposes to use Kt[#], a temporal counting logic, as a formalism for characterizing soft attention transformers. The paper also introduces C-RASP, a variant of RASP corresponding to Kt[#]. The paper proves that Kt[#] formulas can be compiled into transformers, fixed-precision transformers can be compiled into Kt[#], and it offers a tighter lower bound on expressiveness compared to the best prior result (that used a counting logic).

**Questions To Authors:**

- The introduction states that AHATs and UHATs "are not known to be either a subset or superset of standard, soft-attention transformers." What exactly does this mean? It seems to me that AHATs and UHATs clearly aren't supersets of soft-attention transformers. Is the point that some AHATs/UHATs might not be expressible using a standard transformer? Is there a reference for this point?
- Could you elaborate on why exactly FOC[+;MOD] is underpowered (section 2.2)? Is this saying that every sentence in FOC[+;MOD] can be expressed as a two layer transformer? I think it would be helpful to make this more clear.
- Regarding the translation from a transformer to a Kt[#] expression: Would it be possible to give any characterization of the size of the Kt[#] expression?

**Reasons To Accept:**

- This paper is written very clearly for the most part, and I enjoyed reading it.
- I think Kt[#] could be a useful logic for thinking about softmax transformers. Kt[#] brings together several ideas from prior work: counting logics (used by Chiang et al., 2023; Merrill and Sabharwal, 2023) and temporal logics (used by Angluin et al., 2023; Barcelo ́ et al., 2024), and authors also connect it to RASP (Weiss et al., 2021). I find this to be a nice and intuitive formalism for thinking about transformers: temporal logics are a natural way of thinking about operations on strings, and incorporating counting makes them more expressive and better aligned with soft attention/hard averaging attention transformers.
- I also appreciate that the paper provides explicit feed-forward constructions for the unary predicates in the logic. Compared to RASP/Tracr, this means all expressions can be compiled into a standard transformer, and it also makes it possible to give upper bounds on the number of layers.
- The paper gives a tighter lower bound for soft attention transformers.

Overall, despite some reservations (see below), I think the paper is clearly written and introduces an interesting and potentially useful set of tools, and I would recommend accepting it.

**Reasons To Reject:**

- It's not entirely clear to me how much these results reveal about soft attention transformers in particular, as apposed to averaging hard attention transformers (AHATs). It seems that the translation from Kt[#] to transformers (section 5.3/appendix B.2) exclusively uses uniform attention. (So Kt[#] is not necessarily so useful as a language for thinking about soft attention transformers---for example, constructing transformers with soft but non-uniform attention patterns.) It does seem notable that that softmax transformers can be decompiled into Kt[#] expressions (but this is also true of FO[+]). I think it would be helpful to include some discussion about why Kt[#] is specifically useful for softmax transformers and not just AHATs.
- The paper states that it derives a "much better lower-bound result," but I find it a little difficult to understand how much better it is. From what I understand, the main lower-bound result is that Kt[#] can define Dyck-1, but FO[+;MOD] cannot. It would be nice if there was a more general characterization of the lower-bound, or at least a more detailed discussion of how exactly this compares to expressivity results from prior work.
- The paper might benefit from some more discussion of other formalisms in section 6, like first-order logic with majority quantifiers (Merrill and Sabharwal, 2023). In particular, Angluin et al. (2023) also introduce a RASP variant (Boolean RASP) for unique hard attention transformers, and prove that it recognizes exactly the star-free languages. I think it would be worthwhile to discuss the relationship between these results at greater length.

To summarize, my main concerns are related to how the contributions are framed relative to prior work.
While I still think the paper is worth accepting, it would be stronger if it could offer a clearer discussion of how exactly these results differ from prior work.

---

> ### Author Rebuttal · Authors · 2024-05-31
>
> Thank you for your review!
>
> You observed that our translation from Kₜ[#] to transformers uses uniform attention, so Kₜ[#] is also a lower bound on AHATs; we'll add this result to the final version! Then, you ask what our results say about softmax-attention transformers (SMATs) as opposed to AHATs. [Barcelo et al 2024] prove LTL(C,+) is a lower bound on AHATs, but their construction relies on the "winner-take-all" behavior of AHATs, and it's unclear how to adapt it to SMATs. By contrast, Kₜ[#] restricts LTL(C,+) just enough so that it can be simulated by SMATs, while remaining above the previous lower bound of FOC[+;MOD].
>
> Comparisons with other bounds:
>
> - FOC[+;MOD] is "underpowered" and Kₜ[#] is a "much better" lower bound because of the example of Dyck-1, which is relevant for practical applications, and because FOC[+;MOD] translates to 2-layer transformers whereas Kₜ[#] translates to deeper transformers.
>
> - FOM[BIT] ([Merrill and Sabharwal 2023]) is likely a much looser upper bound than Kₜ[#]. For instance, integer multiplication is in FOM[BIT], but transformers are observed to struggle with it ([Dziri et al 2023]), and it does not seem definable in Kₜ[#].
>
> - B-RASP ([Angluin et al 2023]) cannot express Dyck-1, but Kₜ[#] can. On the other hand, FO[<] can express Σ\*aaΣ\*, which we suspect Kₜ[#] and SMATs can't without positional encodings.
>
> # Answers to questions:
>
> 1. We agree UHAT languages are likely a subset of AHAT and SMAT languages. To our knowledge, this has not been shown (we hope to in future work). However, the relationship between AHAT and SMAT is not clear: SMATs have nonuniform attention, but AHATs have "winner-take-all" behavior, suggesting AHATs may recognize a language SMATs can't, or vice versa. We know of no examples, but note that all published proofs that transformers with chain-of-thought are Turing-complete (https://arxiv.org/abs/1901.03429, https://arxiv.org/abs/2006.09286, https://arxiv.org/abs/2310.07923) use average-hard attention.
>
> 2. On FOC[+;MOD], please see above.
>
> 3. Regarding size, we'll note in the final version the formula will have depth O(L) for L attention layers (even with multiple heads), and width $O((2^F)^d)$ in the precision F and width d of the transformer.
>
> [Angluin et al 2023]: https://arxiv.org/abs/2310.13897
> [Barcelo et al 2024]: https://arxiv.org/abs/2310.03817
> [Dziri et al 2023]: https://arxiv.org/abs/2305.18654
> [Merrill and Sabharwal 2023]: https://arxiv.org/abs/2210.02671

---

> > ### Comment · Reviewer_QRfH · 2024-06-04
> >
> > Thank you for the response! I will keep my score as is.

---

> > > ### Author Response · Authors · 2024-06-06
> > >
> > > Thanks very much for your feedback!

---

### Official Review · Reviewer_NmSc · 2024-05-11

**Rating:** 7
**Confidence:** 4
**Ethics Flag:** 1

**Summary:**

This paper introduces two equivalent formalisms (a temporal logic Kt[#] and a RASP fragment called C-RASP) which aim to describe counting capabilities in transformers.
The authors show that they are the tightest known lower bound on the expressive power of transformers with causal masking (no positional embeddings).
Simultaneously, they show that such transformers, when using fixed-precision, can be compiled back into these formalisms.

Given the topic, I found the dimensions listed on https://colmweb.org/ReviewGuide.html very hard to apply. Nonetheless, I believe the paper to be a good fit to the "Science of LMs" Research Area of COLM. I'm evaluating the paper according to the four categories given in the review form:

CLARITY: I found the paper to be generally clearly written and presented. The paper nicely fits with a recent line of research on investigating the expressive power of transformers.

QUALITY: I followed the constructions and believe the formal results to be correct.

ORIGINALITY: My main concern here is the comparison to the recent result about  LTL(C, +) proven by [1], as described under “Questions”. Clarification from the authors could help here.

SIGNIFICANCE: Providing stronger formal lower bounds on the expressive capacity of transformers addresses foundational questions about modern LMs.

**Questions To Authors:**

* I would like to know more about the link to LTL(C, +), proven by [1] to lower-bound AHAT. An obvious difference is that LTL(C, +) includes general unary predicates, establishing that it is strictly more powerful than Kt[#], but this difference just concerns whether or not positional encodings are used. The authors point out that the result from the paper under review applies to soft-attention transformers, different from and more realistic than AHAT. Nonetheless, I’m curious about what’s so different: can’t the construction from [1] be just adapted to softmax transformers?
* How is a finite-precision transformer defined? While the proof of Lemma C.7 asks the reader to recall the definition, I couldn’t find it anywhere. Naively: For any given head, its attention weights will be 1/N on average where N is the input length. When N is large and precision is finite, many of these will – if one naively applies finite precision – be rounded to 0. If one claims finite precision as a realistic theoretical assumption for transformers, then this seems like an undesirable result, as it effectively prevents finite-precision transformers from performing averaging. Lemma C.7 apparently gets around this by applying rounding after division. Is that the *definition* of finite-precision transformers?
* Page 7: “integer values get overwritten” – I didn’t follow this, the statement might be made clearer.
* first paragraph of Section 5.1 has LaTeX "??"


[1] Pablo Barcel ´o, Alexander Kozachinskiy, Anthony Widjaja Lin, and Vladimir Podolskii. Logical languages accepted by transformer encoders with hard attention. In Proc. ICLR, 2024.

**Reasons To Accept:**

* Provides simple and intuitive formalisms lower-bounding the capacity of realistic transformers, addressing foundational questions about transformers and LMs
* Unlike a good deal of the literature on transformer expressiveness, this work applies to realistic soft-attention transformers
* Provides a construction applying at arbitrary input lengths
* Provides some idea of how the result could lead to understanding of expressiveness, through example languages and a construction of piecewise testable languages

**Reasons To Reject:**

Weaknesses:
* Except for the finite precision setting (with sinusoidal embeddings), the results do not apply to transformers with positional embeddings.
* As described in the Summary and under Questions, I believe the paper does not currently do a sufficiently good job of explaining how the result substantially adds over the  LTL(C, +) result by [1]: this appears to be a similar recent result, in that it also shows that a temporal logic lower-bounds the power of a certain formalization of transformers. The paper under review could do a much stronger job making clear how the new results are substantially novel compared to the LTL(C, +) result beyond replacing positional encodings with positional masking and AHAT by soft attention, or why these changes require substantial innovations.
UPDATE: The authors have provided useful clarification, which I encourage them to include in the paper.

---

> ### Author Rebuttal · Authors · 2024-05-31
>
> Thank you for your insightful review. Regarding the lack of position embeddings, we've considered modular predicates, but it is possible, as [Angluin et al., 2023] do for LTL, to consider any finite-image position embedding in Theorem 7.1.
>
> Regarding LTL(C,+), the short answer is that LTL(C,+) is not known to be simulatable by SMATs, but Kₜ[#] restricts LTL(C,+) just enough so that it can be simulated by SMATs, while remaining above the previous bound of FOC[+;MOD]. For the longer answer, please see below.
>
> # Responses to questions
> 1. Indeed, the fact that our results apply to the softmax-attention transformers (SMATs) used in practice rather than AHATs is the main distinction from [Barcelo et al 2024]. Their construction uses a table-lookup operation that relies crucially on the "winner-takes-all" behavior of hard attention, so it cannot directly be applied to SMATs.
> Further differences include:
> - They use nonstandard positional encodings to implement the table-lookup operation. In contrast, ours does not require positional encodings.
> - They interleave unmasked self-attention layers to "broadcast" a final value to every other position. Our construction uses only future-masked attention, like in practice.
> - They do not consider layernorm, so any construction that relies on maximizing an attention score at a particular position, like their Lemma 1 or simulation of X ("next" operator), may break when the position-dependent scaling factor of layernorm is added. In contrast, our construction works with (and indeed requires) layer normalization.
> 2. Indeed, attention weights may become zero on large enough inputs if we round to the nearest number, but we don't think this is a fatal objection. Another reasonable policy, like rounding up, would avoid this. It's also true that changing the order of operations might affect the final output; for example, [Li et al., 2024] argue that rounding should take place after every term of the summation. Our final version will make explicit where and how rounding should take place.
> 3. Regarding the comment on "overwritten" values, further explanation is on page 20. Our construction appends computed values to the "residual stream", where they are left untouched. However, integers eventually get clipped to ±1 due to technicalities of the comparison operation. The correctness of the construction is unaffected.
>
> [Angluin et al., 2023]: https://arxiv.org/abs/2310.13897
> [Li et al., 2024]: https://openreview.net/pdf?id=3EWTEy9MTM

---

> > ### Comment · Reviewer_NmSc · 2024-06-02
> >
> > I thank the authors for the useful response.
> >
> > The points about LTL(C,+) are well taken. I encourage the authors to make the differences more explicit in the revision.
> >
> > > Indeed, attention weights may become zero on large enough inputs if we round to the nearest number, but we don't think this is a fatal objection. Another reasonable policy, like rounding up, would avoid this.
> >
> > If one rounded up values on the order of 1/N to some small nonzero fixed-precision number \epsilon, and then summed up over all inputs, one would get a number on the order \epsilon N simply when each position has the value 1 -- certainly an undesirable situation, and seemingly breaking the proof of Lemma C.7. Perhaps rounding after division leads to more reasonable results. I take the authors' word that they will make this more explicit in the final version.
> >
> > In reaction to the authors' response, I am increasing my score.

---

> > > ### Author Response · Authors · 2024-06-06
> > >
> > > Thanks very much for your feedback and for reconsidering your score!

---

### Official Review · Reviewer_TQ9j · 2024-05-13

**Rating:** 5
**Confidence:** 3
**Ethics Flag:** 1

**Summary:**

The paper studies the representation power of Transformer and proves that constant layer Transformer could recognize all the formal languages defined by formulas of Kt[#]. This is proved by showing Transformer could recognize C-RASP, a variant of RASP (Restricted Access Sequence Processing Language), and then prove the equivalence of Kt[#] and C-RASP.

**Questions To Authors:**

.

**Reasons To Accept:**

The paper shows Transformer could recognize language in Kt[#], this gives some improvement on the previous expressiveness result.

**Reasons To Reject:**

While the construction of Transformer to recognize language in Kt[#] gives some improvement on the previous expressiveness result, the improvement is not so clear since there is no good characterization of Kt[#], and this result is very hard to parse for people outside of programming language area. Moreover, the expressive result is restricting as it does not count the decoding step performed by Transformer (e.g. CoT) and this gap might be very large.


-----------------
Post rebuttal: I slightly raise my score.

---

> ### Author Rebuttal · Authors · 2024-05-30
>
> Thank you very much for your review.
>
> You wrote that our results are an improvement over previous results, but the improvement is not so clear. As a lower bound, Kₜ[#] is better than FOC[+;MOD] in the sense of defining a strict superset of languages, but we think you're asking for some intuition of how much better it is. In short, Kₜ[#] is able to account for the expressive power gained by using causal masking and by increasing depth, which the previous result FOC[+;MOD] cannot.
>
> First, the paper gives the particular example of Dyck-1, which is definable in Kₜ[#] but not FOC[+;MOD]. Many other separating languages exist, due to the inability of FOC[+;MOD] to model ordering, but this is an especially important one. It has clear practical relevance to modeling natural languages and programming languages, and we also know empirically ([Ebrahimi, 2020]) that transformers can learn Dyck-1.
>
> Second, whereas the translation from FOC[+;MOD] outputs a transformer of bounded depth, our translation from Kₜ[#] outputs deeper transformers for deeper formulas. While we don't know whether deeper transformers are in fact more expressive, we suspect that they are, and practical applications certainly suggest that they are. So the fact that Kₜ[#] translates to deeper transformers suggests Kₜ[#] is much stronger and closer to actual transformers than FOC[+;MOD].
>
> You also wrote that our results do not take into account decoding steps, e.g., chain-of-thought. The expressive power of transformer decoders that are allowed to take intermediate steps (chain-of-thought) is by now fairly well understood ([Merrill and Sabharwal, 2024]), and you're right that intermediate steps make the model much more expressive. But there are many situations when it's not desirable to allow intermediate steps; in particular, training data for intermediate steps is not always available. So understanding what transformers are capable of in a single step remains an important question.
>
> [Ebrahimi, 2020]: https://aclanthology.org/2020.findings-emnlp.384/
> [Merrill and Sabharwal, 2024]: https://arxiv.org/abs/2310.07923

---

> > ### Comment · Reviewer_TQ9j · 2024-06-03
> > **Thank you for your reply**
> >
> > > First, the paper gives the particular example of Dyck-1, which is definable in Kₜ[#] but not FOC[+;MOD]. Many other separating languages exist, due to the inability of FOC[+;MOD] to model ordering, but this is an especially important one. It has clear practical relevance to modeling natural languages and programming languages, and we also know empirically (Ebrahimi, 2020) that transformers can learn Dyck-1.
> >
> > The inclusion of Dyck1 is nice. Do you have other examples that are not in FOC[+; MOD] but in $K_t[#]$? I ask this because [1] already shows Dyck can be represented by two-layer Transformer. (actually not only dyck1 but also general deck language, but of course it only shows dyck so it has no confliction with this work)
> >
> > [1]  Self-attention networks can process bounded hierarchical languages, ACL 2021, Yao S, Peng B, Papadimitriou C, et al.
> >
> > > Second, whereas the translation from FOC[+;MOD] outputs a transformer of bounded depth, our translation from Kₜ[#] outputs deeper transformers for deeper formulas. While we don't know whether deeper transformers are in fact more expressive, we suspect that they are, and practical applications certainly suggest that they are. So the fact that Kₜ[#] translates to deeper transformers suggests Kₜ[#] is much stronger and closer to actual transformers than FOC[+;MOD].
> >
> > What do you mean by deeper? Does the number of layer you constructed depends on $t$?

---

> > > ### Author Response · Authors · 2024-06-04
> > > **Response to reviewer**
> > >
> > > We appreciate you taking a second look. See our responses below:
> > >
> > > 1. Other examples of languages definable in Kₜ[#] but not FOC[+;MOD] include:
> > >
> > > - $a^nb^nc^n$ (§3.2), a context-sensitive language  which is also observed to be learnable by [Bhattamishra, 2020].
> > >
> > > - All piecewise testable languages, like $\Sigma^\*a\Sigma^\*b\Sigma^\*c\Sigma^\*$ (§3.3).
> > >
> > > - All strings with the letter $a$ in the second position and nowhere else.
> > >
> > > - All valid email addresses (name@domain.com)
> > >
> > > We make no claim of novelty in showing that Kₜ[#] and therefore transformers can recognize Dyck-1. The paper already cites [Bhattamishra et al, 2020] for showing that transformers recognize a language similar to Dyck-1, and we thank you for pointing out [Yao et al, 2021], which is a better reference.
> > >
> > > Our focus here is not just on individual languages, but on the formalisms Kₜ[#] and C-RASP, which provide building blocks that can be combined to show that transformers recognize a whole class of languages.  The fact that [Yao et al, 2021] shows that transformers can recognize Dyck-1 strengthens, rather than weakens, our claim that Kₜ[#] is a better characterization of transformers than FOC[+;MOD].
> > >
> > > At present, we don't know how to define Dyck-2 in Kₜ[#], so this would be a good target for future work. But we note that [Yao et al, 2021]'s Theorem B.3 appears to use hard attention, and also relies on Lemma B.1, which implicitly uses a nonstandard variant of layer normalization that can apply only to a subset of vector components (cf. multi-pre-norm from [Merrill and Sabharwal, 2024]). As such, it may not be straightforward to apply their construction directly to standard transformers, which is the setting we are working in.
> > >
> > > 2. By "deeper," we mean the number of transformer blocks (where a block is self-attention layer followed by a feed-forward layer). Our construction simulates a Kₜ[#] formula that has modal depth $k$ (§3.3) by a transformer with $4k+1$ blocks. (We are unsure what you mean by $t$, but to be clear, the depth of the transformer does not depend on the input string length.)
> > >
> > > This is in contrast to FOC[+;MOD], which [Chiang et al, 2023] show can be simulated by a transformer that only needs $2$ blocks. Since in practice it seems that deeper transformers are better for more complex applications, it seems to be a strength of Kₜ[#] that it translates into deeper transformers.
> > >
> > > Let us know if you have more questions or concerns, and we would be glad to elaborate!
> > >
> > > [Bhattamishra et al, 2020]: https://aclanthology.org/2020.emnlp-main.576
> > >
> > > [Chiang et al, 2023]:
> > > https://arxiv.org/abs/2301.10743
> > >
> > > [Merrill and Sabharwal, 2024]: https://arxiv.org/pdf/2310.07923
> > >
> > > [Yao et al, 2021]:
> > > https://aclanthology.org/2021.acl-long.292/

---

> > > > ### Author Response · Authors · 2024-06-05
> > > > **Checking for additional concerns**
> > > >
> > > > We thank the reviewer for the valuable comments. Since it is close to the end of author-reviewer discussion, please let us know if we have properly addressed your questions.

---

> > > > > ### Comment · Reviewer_TQ9j · 2024-06-07
> > > > >
> > > > > I thank the authors for the rebuttal. I slightly raise my score. I am still not very excited by the improvement, but I will carefully read other reviews and ask for their opinions.

---

### Decision · Program_Chairs · 2024-07-10

**Decision:**

Accept

**Comment:**

The paper studies the problem of characterizing the classes of computational problems that can be solved by autoregressive transformers. This is done purely from the expressive power perspective. This is an important theoretical problem with considerable recent interest. The paper under review provides a new class of problems that can be expressed in a temporal counting logic. They also provide another equivalent formalism C-RASP. Their formalism is natural and clean and arguably provides the largest class of computational problems expressible by transformers so far. There are technical issues here that arise when comparing to prior work regarding soft and hard attention and other elements of transformers. The current paper appears to be closer to transformers as used in practice.

Out of the four, three reviewers were positive about the paper. Among the questions, perhaps the most important was about how much bigger is the class of problems proposed in the present paper compared to the previously known classes. This remains an open question. Nevertheless, I think the contributions of the present paper are sufficient to warrant acceptance.